# The complexity of the population dynamics of *Triatoma brasiliensis* in rural north-east Brazil indicated by genetic characterisation

Luiz Osvaldo Rodrigues Silva[1], Carlota Josefovicz Belisário[2], Flávio CamposFerreira[2], Jorg Heukelbach[1], Liléia Diotaiuti[2]/+, Claudia Mendonça Bezerra[3]

[1]Universidade Federal do Ceará, Fortaleza, CE, Brasil
[2]Fundação Oswaldo Cruz-Fiocruz, Instituto René Rachou, Belo Horizonte, MG, Brasil
[3]Secretaria Estadual da Saúde do Ceará, Fortaleza, CE, Brasil

**BACKGROUND** *Triatoma brasiliensis*, the primary Chagas disease (CD) vector in the north-east of Brazil, poses a significant challenge for control due to its adaptability and ability to colonise anthropic environments. The limited number of previous studies on the population dynamics of *T. brasiliensis* hinders the development of effective control strategies.

**OBJECTIVES** This study characterises the genetic variability of *T. brasiliensis* populations in Jaguaruana using microsatellite markers, in order to understand the population processes of triatomine infestation and reinfestation.

**METHODS** We analysed the genetic structure of 229 *T. brasiliensis* specimens collected in the municipality of Jaguaruana in the north-east Brazilian State of Ceará using microsatellite markers.

**FINDINGS** Hardy-Weinberg disequilibrium prevailed, with substantial genetic variability (67.2%) among individuals and inbreeding, but genetic differentiation lacked correlation with geographical distance (Mantel's test).

**MAIN CONCLUSIONS** The complex population dynamics in Jaguaruana revealed diverse sources of anthropogenic colonisation, impacting regional control. This study underscores the necessity of comprehending intricate infestation processes for planning effective vector surveillance and control strategies.

Key words: Chagas disease - Triatominae - *Triatoma brasiliensis* - microsatellites - genetic variability - Ceará

Chagas disease (CD) or American trypanosomiasis is caused by the protozoan parasite *Trypanosoma cruzi*, which infects human and non-human animal hosts, through oral transmission, contact with the infected faeces of triatomines and vertical transmission from mother-to-offspring. According to the World Health Organization (WHO), CD is considered a neglected tropical disease.[1] CD is endemic in 21 countries in the Americas, and it is estimated that around 6 million people worldwide are infected with *T. cruzi*, with 75 million people living in areas at risk of infection.[1] In Brazil, there are between 1.9 and 4.6 million people infected, with the majority suffering from the chronic form of CD.[2,3]

In Brazil, 64 species of triatomines have been recorded.[4] Of these, 23% (15) are present in the North-East region of the country, with four species found predominantly inside human dwellings, highlighting the ability of these vectors to establish colonies in human habitations.[5,6,7]

*Triatoma brasiliensis brasiliensis* (Neiva, 1911) is the most important vector in the Brazilian North-East region.[8,9,10,11] *Triatoma b. brasiliensis* is a rupestrian sub-species, and in its natural ecotope is mainly associated with rodents, marsupials, and bats.[10,12,13] This triatomine subspecies is capable of invading and colonising domiciles and diverse peridomiciliary ecotopes. *Triatoma b. brasiliensis* exhibits a highly eclectic diet, is aggressive, opportunistic, and has significant rates of *T. cruzi* infection. These characteristics make *Triatoma b. brasiliensis* the primary vector of *T. cruzi* transmission within the Caatinga region of the north-east of Brazil.[9,14,15]

In the municipality of Jaguaruana in the State of Ceará, CD vector control was implemented in the 1970s. About two decades ago, a seroprevalence study[16] revealed a seropositivity rate of 3.1%, including children under 10 years of age and patients with cardiovascular or digestive symptoms. *T. cruzi*-infected triatomines have also been found in rural localities of Jaguaruana.[17] Today, in this region, there are still significant home infestations with triatomines, which may be related to several factors previously described in the literature, such as the varied sources of wild and peridomiciliary infestation, the complexity of domestic shelters, opera-

**doi:** 10.1590/0074-02760250076
**Financial support:** Instituto René Rachou/Fiocruz Minas, Coordination of Health Surveillance, Fiocruz Reference Laboratories and the Department of Health of Ceará State (SESA-CE).
LORS and CJB contributed equally to this work.
+ **Corresponding author:** lileia.diotaiuti@fiocruz.br | ⦿ https://orcid.org/0000-0002-4976-2618

**Handling editor:** Adeilton Alves Brandão | ⦿ https://orcid.org/0000-0001-5877-607X

tional failures in chemical vector control activities, and the resistance of triatomine populations to the insecticides used in vector control.[8,10,11,12]

In Tauá, another municipality in the State of Ceará, with a different natural ecotope from Jaguaruana (Fig. 1), a previous population genetic study using microsatellites was able to show that *T. brasiliensis* had a panmictic population structure, and, therefore, required intense entomological surveillance in order to control the early reestablishment of infestation foci after the application of insecticide control measures.[18] In the State of Paraíba, also located in north-eastern Brazil, Almeida et al.[19] used the mitochondrial *cytb* gene and, in contrast to the study described above, suggested that *T. brasiliensis* populations are genetically structured. Furthermore, these authors observed that reinfestation of anthropic and/or disturbed natural environments is by triatomine individuals from distinct populations. Another study using microsatellites conducted in the State of Rio Grande

do Norte, also in north-eastern Brazil, demonstrated gene flow between the distinct populations of *T. brasiliensis* found in sylvatic environments and those from anthropic and/or disturbed natural ecotopes.[20] Yet another study, undertaken in Currais Novos, a municipality in Rio Grande do Norte, where sequencing of the *cytb* gene revealed four mitochondrial clusters within the 13 sampled *T. brasiliensis* populations. In the same study, analysis of single nucleotide polymorphisms (SNPs) indicated, at most, only very low levels of population genetic structuring suggestive of very high levels of gene flow, if not panmixia.[21]

In general, studies on the variability of microsatellite alleles are useful for understanding population genetic structure, taxonomy, and genomic mapping. Such studies can provide information on gene flow between populations, vector dispersal, and the taxonomic assessment of vectors.[20-33] Microsatellite markers have been shown to be a good tool for investigating the dynamics of triatomine populations, and the design and implementation of new vector control strategies.[20-33] Knowledge about genetic processes and gene flow between environments can elucidate the process of infestation and reinfestation of households by autochthonous triatomines, such as *T. brasiliensis*, which is adapted to both sylvatic and domestic environments.[18,20]

This current study characterises the genetic variability of *T. brasiliensis* populations in Jaguaruana using microsatellite markers, in order to evaluate the extent of gene flow between triatomines in this region and further demonstrate that such population genetic analysis is a useful tool for understanding the population processes of triatomine infestation and reinfestation.

## MATERIALS AND METHODS

*Study area* - This study was conducted in the municipality of Jaguaruana, an arid region in the Caatinga in the State of Ceará in the North-East region of Brazil (Fig. 2). Jaguaruana is in the Jaguaribe mesoregion (4º50'02"S; 37º46'52"W), at an altitude of 20 metres above sea level (asl) and 150 km from the capital city of the state, Fortaleza. The climate is mild semi-arid warm tropical, with an average temperature between 26ºC and 28ºC, an average rainfall of 753 mm³, and a rainy season from January to April.[33] The study area has a mix of vast carnaúba palm forests [*Copernicia prunifera* (Mill.) H. E. Moore] and desert areas with xerophytic vegetation, shrubby and spiny, where the xique-xique cactus (*Pilosocereus gounellei*, F.A.C. Weber) is abundant and commonly serves as a shelter for small rodents and reptiles.[34]

*Triatoma brasiliensis* is found in practically the entire municipality, both in sylvatic and anthropic environments. The present study included rural locations with a history of *T. brasiliensis* infestation, which also regularly implemented insecticide-based vector control, namely: Latadas [44 domiciliary units (DUs)], Cipriano Lopes (36 DUs), Jenipapeiro (36 DUs) and Quixabinha (28 DUs) (Fig. 2). A DU usually consists of both the intradomicile (human habitation) and the peridomicile (surroundings of the human dwelling), including fences, animal shelters, piles of objects (tiles, bricks, stones,

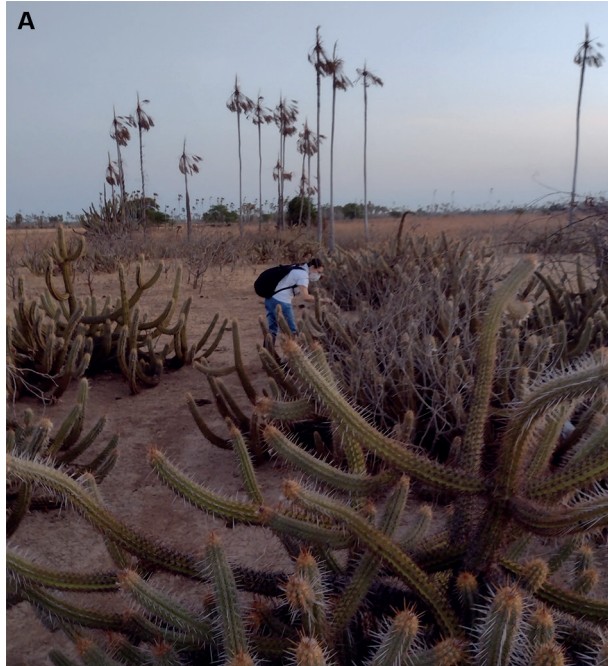

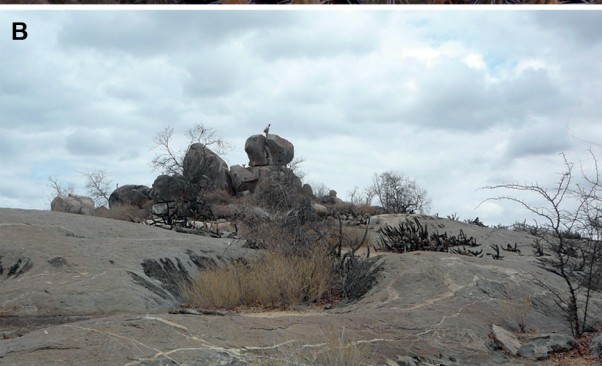

Fig. 1. main sylvatic ecotopes of *Triatoma brasiliensis* in Ceará State. (A) In Jaguaruana municipality, the cactus *Pilosocereus gounellei*, which is distributed in discontinuous clumps. (B) In Tauá municipality, large and extensive granite formations.

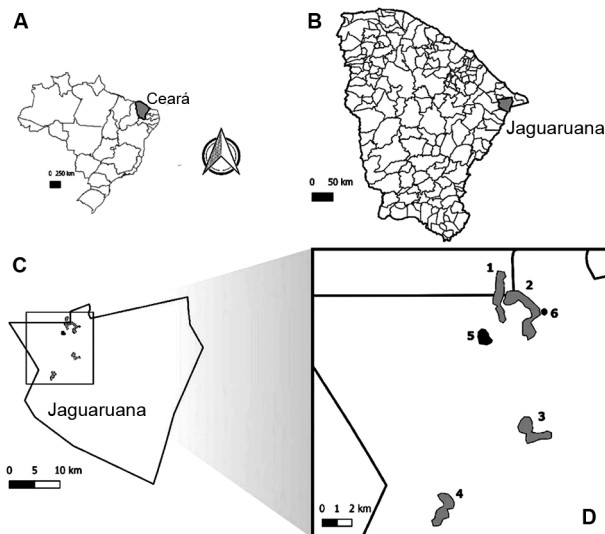

Fig. 2: study area. (A) State of Ceará, Brazil; (B) Municipality of Jaguaruana; (C) Localities included in Jaguaruana; (D) Study localities. Gray polygons: areas with collection on compounds: 1: Latadas; 2: Cipriano Lopes; 3: Jenipapeiro; 4: Quixabinha; black polygons: areas with sylvatic collection; 5: João Duarte; 6: Cipriano Lopes.

wood, etc.), as well as permanent and temporary constructions. The ecological complexity and stability of the peridomestic ecotope is responsible for maintaining populations of triatomines, where availability of shelter, food sources, and relatively constant abiotic conditions favour colonisation and high population densities. Sylvatic vector samples were also collected from five rocky outcroppings in the locality of João Duarte, and from a cluster of xique-xique in Cipriano Lopes (Fig. 2).

*Collection of Triatomine samples* - The triatomines were captured manually and exhaustively in the DUs by endemic disease agents from the municipality of Jaguaruana, following the recommended standard procedures,[35,36] in locations monitored by the State Health Department in October 2021. DUs with the presence of triatomines (intradomiciliary and/or peridomiciliary) were sprayed with alpha-cypermethrin SC 20% (Fersol Industria e Comércio). The triatomines collected were identified according to DU and ecotope of origin.

For the population genetic analysis, 252 insects were used, pooled into 29 separate samples. We also analysed other frozen triatomines that were collected in the same locations between 2016 and 2018 and were provided by REMOT (Monitoring Network for the Susceptibility of Brazilian Triatomine Populations to Insecticides). The individual triatomines were pooled into samples according to their collection site (*i.e.*, individual DU or sylvatic environment). When triatomines were collected from more than one ecotope within the same DU, separate pooled samples were made for each different ecotope, while triatomines captured from different sites within the same sylvatic location were also pooled separately (Table I). Each pooled sample was composed of a minimum of five individuals as required for analysis of molecular variance (AMOVA).[36,37]

*Microsatellite genotyping* - Two legs from each insect were used for genomic DNA extraction using the Wizard® Genomic DNA Purification Kit (Promega) and the protocol of Borges et al.[10] The DNA was quantified using a NanoDrop 1000 Spectrophotometer (Thermo Scientific) and stored at -20ºC. Primers were tested for nine microsatellite loci for *T. brasiliensis*: Tb728, Tb830, Tb860, Tb7180, Tb8112, Tb8124,[38] B2146 (GenBank: KT355796.1), B8102 (GenBank: KT355797.1) and B8150 (GenBank KT355795.1).

Polymerase chain reactions (PCR) amplifications were carried out using a final volume of 10 µL containing 1 unit of Platinum® Taq DNA polymerase (Invitrogen), 1X buffer, 1.5 mM MgCl$_2$, 1 mM dNTP, 5 pmol for each primer, 2 ng of DNA and ultrapure water. The forward primers were labelled with a bioluminescent probe. The reactions were performed using a Veriti® 96-Well thermal cycler (Thermo Fisher Scientific) with the following cycling conditions: an initial denaturation at 95ºC for 5 min; followed by 28 cycles of 94ºC for 30 s, primer-dependent temperature annealing for 30 s, and extension at 72ºC for 45 s; and then a final extension at 72ºC for 5 min. The annealing temperatures for each locus were: 48ºC for Tb860; 54ºC for Tb8112; 52ºC for Tb 2146; and 56ºC for Tb8102; followed by touchdown (*i.e.*, incremental reduction in the annealing temperature): 60→50ºC and 58ºC for Tb728, Tb830, Tb7180, Tb8124. In order to determine the size of the amplicons, the PCR products were diluted 1:10 in pure water together with a GeneScan™ 500 LIZ® size standard (Thermo Fisher Scientific) and genotyped on an ABI 3730 DNA Analyzer (Applied Biosystem®) at the DNA Sequencing Platform of the René Rachou Institute. The chromatograms were analysed using the Geneious 10.1.2© program (Biomatters Limited).

*Data analysis* - Several analyses were conducted: obtaining the number and size of alleles for each locus, observed (OH) and expected heterozygosity (EH), Hardy-Weinberg equilibrium (HW) verification, AMOVA, calculation of fixation indices ($F_{st}$, $F_{is}$, and $F_{it}$) and the Mantel test (Arlequin version 3.5).[39] The statistical tests were carried out using a significance level of 5% and a maximum loss of 5% of amplified alleles. Due to the large number of pairwise $F_{st}$ comparisons, p-values were corrected using the false discovery rate (FDR) method to control the false positive rate.[40] A Neighbor-Joining dendrogram was generated with the pairwise $F_{st}$ values using POPTREEW.[41] It is important to emphasise that the primary purpose of the tree is to make it easier to visualise the relationships obtained by the pairwise $F_{st}$. Only bootstrap values greater than 50% are statistically supported and were arbitrarily rooted because no outgroup was used.

Hardy-Weinberg equilibrium deviations were evaluated in Genepop v4.3 using the Guo and Ye exact test. The Markov chain procedure was conducted with 10,000 steps, 20 independent replicates, and 5,000 iterationsper replicate.[42,43]

Allelic richness was calculated using the rarefaction statistical method based on a minimum of ight genes per sample (HP-Rare version 1.1).[44,45]

TABLE I

Number of *Triatoma brasiliensis* insects captured, by location, ecotope and year
in the municipality of Jaguaruana in the State of Ceará, Brazil

| Location of collection | Sample abbreviation | Domiciliary unit | Intra/Peri/ Sylvatic | Ecotope | Year of collection | Southern latitude | West longitude | Number of insects |
|---|---|---|---|---|---|---|---|---|
| Latada | Lat18c1 | NI | Peri | NI | 2016 | -4.7738593 | -37.8306312 | 9 |
| | Lat23 | NI | Peri | NI | 2018 | -4.7738593 | -37.8306312 | 10 |
| | Lat3c1 | 18c1 | Peri | chicken coop | 2021 | -4,7637007 | -37,831751 | 10 |
| | Lat11 | 23 | Peri | chicken coop | 2021 | -4,7599111 | -37,8308041 | 10 |
| | Lat13 | 3c1 | Peri | roof tile | 2021 | -4.7065176 | -37.832823 | 10 |
| | Lat14c1 | 13 | Peri | wood | 2021 | -4,7652177 | -37,8314035 | 10 |
| | CLop17 | 14c1 | Peri | chicken coop | 2021 | -4.7646038 | -37.8331039 | 9 |
| | CLop33c1p1 | 11 | Intra | front porch | 2021 | -4.7658002 | -37.832582 | 10 |
| Cipriano Lopes | CLop33c1p2 | NI | Peri | NI | 2016 | -4.78018 | -37.81823 | 10 |
| | CLop15c2 | NI | Peri | NI | 2018 | -4.78018 | -37.81823 | 10 |
| | CLop27 | 33c1 | Peri | firewood 1 | 2021 | -4,7657745 | -37,8271903 | 10 |
| | CLop23p1 | | Peri | firewood 2 | 2021 | -4,7657607 | -37,827158 | 10 |
| | CLop23p2 | 15c2 | Peri | pigsty | 2021 | -4.7764018 | -37.8202555 | 10 |
| | Quix5 | 27 | Peri | pigsty | 2021 | -4.7651053 | -37.8252401 | 10 |
| | Jen6 | 23 | Peri | wood | 2021 | -4.7650495 | -37.8222092 | 9 |
| | Jen1 | | Peri | chicken coop | 2021 | -4.7651269 | -37.8223627 | 10 |
| | Jen6c1 | 17 | Intra | front porch | 2021 | -4.7707044 | -37.8161672 | 7 |
| | Jen15 | NI | Sylvatic | carnaúba-xique-xique-floor | 2021 | -4.7709308 | -37.8157584 | 7 |
| Quixabinha | JDuaWild1 | NI | Peri | NI | 2016 | -4.8555173 | -37.8540889 | 10 |
| | JDuaWild2 | 5 | Peri | chicken coop | 2021 | -4.8536332 | -37.8554654 | 5 |
| Jenipapeiro | JDuaWild3 | 6 | Peri | chicken coop | 2021 | -4.82338 | -37.8188 | 7 |
| | JDuaWild4 | 1 | Peri | carnaúba | 2021 | -4,8259575 | -37,8100437 | 6 |
| | CLopWild | 6c1 | Peri | chicken coop | 2021 | -4,8233762 | -37,81883 | 5 |
| | JDuaWild5 | 15 | Peri | pigsty | 2021 | -4,8227348 | -37,8209339 | 9 |
| João Duarte | LatR25 | - | Sylvatic | rock 3 | 2021 | -4,7827741 | -37,8389039 | 11 |
| | LatR70 | - | Sylvatic | rock 4 | 2021 | -4,7828295 | -37,8393717 | 8 |
| | CLopR26 | - | Sylvatic | rock 5 | 2021 | -4,7825214 | -37,8392454 | 5 |
| | QuixR27 | - | Sylvatic | rock 6 | 2021 | -4,7821158 | -37,839741 | 6 |
| | CLopR69 | - | Sylvatic | rock Luiz | 2021 | -4,7811206 | -37,8386576 | 9 |

NI: Not identified.

The presence of null alleles was checked (Micro-Checker 2.2.3)[46] and their influence was assessed using the null allelic exclusion (NAE) methodology with a 95% confidence interval (CI) generated using 10,000 bootstraps (FreeNa).[47]

The first-generation migrant test was carried out to identify potential immigrants within each sample and their most likely origin.[48] The test was performed using the computational criterion of the frequency-based method proposed by Paetkau[49] together with the algorithm described by Paetkau.[48] This analysis was carried out using a total of 10,000 Monte Carlo chains for each individual simulation with a $p \leq 0.05$ for each result generated (GENECLASS2).[50]

In order to study the population genetic structure of the *T. brasiliensis* sampled, one to 15 genetic clusters ($K$) were evaluated using a total of 20 repetitions and 1,000,000 iterations per Markov and Monte Carlo Chain (burn-in 100,000) for each K evaluated with the correlated allele frequencies between populations (Structure version 2.3.4).[51,52] The best $K$ value was identified using the metrics MedMedK, MedMeanK, MaxMedK, and MaxMeanK[53] ranging from 1 to 10 (Structure Selector[54]).

*Ethics* - This study was approved by the Human Research Ethics Committee of the Federal University of Ceará (UFC), under license no. 6.024.559 on 26 April 2023.

**RESULTS**

Of the nine primer pairs tested, only seven had amplifications, and only five of these amplified microsatellite loci were polymorphic (Table II). For the

TABLE II

*Triatoma brasiliensis* microsatellite loci amplified from samples captured form different rural localities
in the municipality of Jaguaruana in the State of Ceará, Brazil

| Name | | Sequence (5'- 3') | Motif | Annealing temperature | Expected size[38] | Observed size |
|---|---|---|---|---|---|---|
| Tb728 | F: | CTACAGCGATTTGTCTCG-NED | $(GT)_2AT(GT)_{12}$ | 58ºC | 306 - 316 | 308 - 316 |
| | R: | TATTGCATCATGTTTATTGG | | | | |
| Tb830 | F: | TGTCAGATGCATGGTGATAC-6FAM | $(AC)_{15}$ | 58ºC | 274 - 298 | 276 - 290 |
| | R: | CATGGAAGATACCTAAACGG | | | | |
| Tb860 | F: | CGTTTTAGTAAGGAATGG-PET | $(CT)_5 (CA)10(CTCA)_3$ | 48ºC | 392 - 396 | 394 |
| | R: | ATTGTGCCAAAATCAGGT | | | | |
| Tb7180 | F: | TGACCTACCGCCACATTAC-VIC | $(CATA)_3(CA)_8 TA(CA)_{18}(GA)_3$ | 58ºC | 220 - 246 | 214 - 246 |
| | R: | CAAATTTTCGATACCGCGATAG | | | | |
| Tb8124 | F: | GCCACTGTGTTCTCATTCC-NED | $(CA)_{18}$ | 58ºC | 218 - 246 | 224 - 242 |
| | R: | TGGTGTGATGCTCAGAAGG | | | | |

five polymorphic microsatellite loci, the average number of alleles (NA) observed per locus ranged from 1.9 (Tb860) to 5.1 (Tb7180), with an overall average of 3.2. Sample 1 from João Duarte Silvestre (JDuaWild1) had both the highest average allelic number (NA = 4.2) and allelic richness (AR = 3.3), while DU 18c1 from Latada (Lat18c1) had the lowest average NA (= 2.0) and AR (= 1.8). The NA and AR of all the samples are shown in Table III. The size of each individual allele is given in Supplementary data (Table I).

The locus with the lowest average OH was Tb8124 (0.16), and the highest was Tb7180 (0.56). As for the average EH, Tb860 had the lowest average (0.31), and Tb8124 had the highest (0.65), respectively. Loci Tb728 and Tb830 were in Hardy-Weinberg equilibrium (HW). Regarding populations, most showed HW disequilibrium due to an excess of homozygotes (p-values ≤ 0.05 for the heterozygote deficit test), except for CLopR69. The results of OH, EH and HW for the samples are detailed in Table IV.

AMOVA showed that 67.2% of the genetic variability is among all individuals analysed, 22.6% among individuals from the same sample and 10.2% among samples. The fixation indices showed a significant p-value ≤ 0.05 (Table V).

The inbreeding coefficient ($F_{is}$) ranged from -0.09 (Jenipapeiro DU 1 and Cipriano Lopes DU 26 from REMOT) to 0.48 CLopR69. Positive $F_{is}$ values were observed in the following samples: Latadas DUs 23, 3c1, 11; Cipriano Lopes DUs 17, 33c1p1, 15c2, and sylvatic habitat (CLopWild); Jenipapeiro DU 6c1; João Duarte sylvatic habitats 2 and 3 (JDuaWild2 and JDuaWild3). The samples Quixabinha DU 5, Cipriano Lopes 26 from REMOT and Jenipapeiro DUs 1 showed negative $F_{st}$ values. The population differentiation index (pairwise $F_{st}$) ranged from 0 to 0.44. The comparisons with the lowest values were: Cipriano Lopes wild (CLopWild) with Latadas DU 14c1; João Duarte wild environments 2 and 5 (JDuaWild2 and JDuaWild5); Quixabinha DU27 from

REMOT with João Duarte wild ecotope 4 (JDuaWild4); Latadas DU25 and DU70 from REMOT; and Latadas DU 70 from REMOT with João Duarte wild 5 (JDuaWild5). The most differentiated samples were Jenipapeiro DU 6c1 and Latadas DU 18c1. Negative $F_{st}$ indices were considered indicative of no genetic differentiation (Table VI). Mantel's test did not indicate a correlation between genetic differentiation and geographical distance.

The genetic structure of the analysed populations was assessed using a neighbour-joining (NJ) tree based on the pairwise $F_{st}$ index (Fig. 3) and a Bayesian clustering analysis performed using STRUCTURE (Fig. 4). In the dendrogram (Fig. 3), the sylvatic populations of João Duarte (JDuaWild2 and JDuaWild4) formed a single cluster, while the other populations from the same locality did not cluster with the first two. There is also genetic similarity between the populations Latadas REMOT (LatR70) and Quixabinha REMOT (QuixR27). The Latadas population (Lat18c1) was the most differentiated. The dendrogram showed the two peridomiciliary annexes of Cristiano Lopes' DU 23 (CLop23p1 and CLop23p2) clustered. A different situation occurred in the peridomiciliary annexes of DU 33 in the same locality (CLop33p1 and CLop33p2). Three of the four statistics evaluated supported supported K = 6 as the most likely number of clusters [Supplementary data (Figure)]. The genetic structure analysis indicated clusters with high diversity in 18 of the 20 runs performed. The most homogeneous samples were those from Latadas DU 18c1 and DU 23, which did not resemble each other.

The presence of null alleles was observed for each of the five polymorphic loci. However, this did not influence the $F_{st}$ analyses, since the either excluding null alleles (0.105) or including them (0.102) were within the confidence interval of the NAE method (that excludes null alleles) (0.088 to 0.124 without null alleles; 0.091 to 0.118 with null alleles) [Supplementary data (Tables II-IV)].

The test of the first generation of migrants detected 19 individuals, which are shown in Table VI. Quix-

TABLE III

Number of alleles (NA) and allelic richness (AR) per microsatellite locus for *Triatoma brasiliensis* samples collected from the municipality of Jaguaruana in the State of Ceará, Brazil

| Sample\Locus | Tb728 | | Tb830 | | Tb860 | | Tb7180 | | Tb8124 | | Average | |
|---|---|---|---|---|---|---|---|---|---|---|---|---|
| | NA | AR | NA | AR | NA | AR | NA | AR | NA | AR | NA | AR |
| Lat18c1 | 2 | 1.8 | 1 | 1.0 | 2 | 2.0 | 3 | 2.1 | 2 | 2.0 | 2.0 | 1.8 |
| Lat23 | 2 | 1.9 | 3 | 2.7 | 1 | 1.0 | 5 | 3.3 | 2 | 1.6 | 2.6 | 2.1 |
| Lat3c1 | 2 | 2.0 | 3 | 2.4 | 2 | 1.6 | 5 | 4.0 | 5 | 3.8 | 3.4 | 2.7 |
| Lat11 | 4 | 3.5 | 3 | 2.4 | 2 | 1.4 | 4 | 3.2 | 5 | 4.0 | 3.6 | 2.9 |
| Lat13 | 3 | 2.0 | 4 | 3.0 | 2 | 1.9 | 5 | 3.6 | 4 | 3.0 | 3.6 | 2.7 |
| Lat14c1 | 4 | 3.4 | 3 | 2.6 | 2 | 1.9 | 5 | 4.0 | 5 | 3.9 | 3.8 | 3.2 |
| CLop17 | 3 | 2.4 | 3 | 2.8 | 2 | 2.0 | 4 | 3.5 | 4 | 3.2 | 3.2 | 2.8 |
| CLop33c1p1 | 2 | 1.9 | 2 | 1.9 | 2 | 2.0 | 5 | 3.9 | 2 | 1.6 | 2.6 | 2.3 |
| CLop33c1p2 | 4 | 2.7 | 3 | 2.6 | 2 | 1.8 | 6 | 4.2 | 4 | 3.0 | 3.8 | 2.8 |
| CLop15c2 | 3 | 2.8 | 2 | 1.9 | 2 | 1.4 | 6 | 4.0 | 2 | 1.9 | 3.0 | 2.4 |
| CLop27 | 3 | 2.8 | 4 | 2.9 | 2 | 1.9 | 6 | 4.4 | 4 | 3.0 | 3.8 | 3.0 |
| CLop23p1 | 2 | 1.8 | 4 | 3.4 | 2 | 1.8 | 6 | 4.4 | 3 | 2.6 | 3.4 | 2.8 |
| CLop23p2 | 2 | 1.4 | 4 | 3.5 | 1 | 1.0 | 6 | 3.8 | 4 | 3.2 | 3.4 | 2.6 |
| Quix5 | 2 | 2.0 | 3 | 3.0 | 1 | 1.0 | 3 | 3.0 | 5 | 4.6 | 2.8 | 2.7 |
| Jen6 | 2 | 2.0 | 2 | 2.0 | 2 | 2.0 | 3 | 2.1 | 3 | 2.7 | 2.4 | 2.2 |
| Jen1 | 2 | 2.0 | 3 | 2.6 | 2 | 2.0 | 5 | 4.2 | 2 | 2.0 | 2.8 | 2.6 |
| Jen6c1 | 2 | 2.0 | 2 | 2.0 | 2 | 2.0 | 4 | 3.8 | 1 | 1.0 | 2.2 | 2.2 |
| Jen15 | 3 | 2.4 | 3 | 2.7 | 2 | 2.0 | 6 | 4.0 | 3 | 2.4 | 3.4 | 2.7 |
| JDuaWild1 | 4 | 3.7 | 3 | 2.9 | 3 | 2.2 | 6 | 4.1 | 5 | 3.4 | 4.2 | 3.3 |
| JDuaWild2 | 4 | 3.4 | 2 | 2.0 | 1 | 1.0 | 7 | 5.2 | 3 | 2.9 | 3.4 | 2.9 |
| JDuaWild3 | 4 | 3.6 | 2 | 2.0 | 2 | 2.0 | 5 | 4.6 | 2 | 2.0 | 3.0 | 2.8 |
| JDuaWild4 | 3 | 2.9 | 2 | 2.0 | 3 | 2.3 | 5 | 4.6 | 4 | 3.9 | 3.4 | 3.1 |
| CLopWild | 3 | 2.4 | 2 | 1.9 | 2 | 2.0 | 5 | 3.8 | 3 | 2.8 | 3.0 | 2.6 |
| JDuaWild5 | 4 | 3.5 | 3 | 2.4 | 2 | 1.6 | 7 | 5.2 | 3 | 2.7 | 3.8 | 3.1 |
| LatR25 | 3 | 2.4 | 3 | 2.8 | 2 | 1.8 | 6 | 4.4 | 3 | 2.9 | 3.4 | 2.9 |
| LatR70 | 3 | 2.2 | 3 | 2.4 | 2 | 2.0 | 5 | 3.9 | 2 | 2.0 | 3.0 | 2.5 |
| CLopR26 | 2 | 1.9 | 2 | 1.8 | 2 | 1.9 | 4 | 3.0 | 4 | 2.9 | 2.8 | 2.3 |
| QuixR27 | 3 | 2.2 | 2 | 2.0 | 2 | 1.9 | 6 | 4.4 | 4 | 3.0 | 3.4 | 2.7 |
| CLopR69 | 3 | 2.5 | 3 | 2.5 | 1 | 1.0 | 6 | 4.9 | 4 | 3.2 | 3.4 | 2.8 |
| Average | 2.9 | 2.5 | 2.7 | 2.4 | 1.9 | 1.7 | 5.1 | 3.9 | 3.3 | 2.8 | 3.2 | 2.7 |
| Total NA | 4 | | 4 | | 3 | | 14 | | 11 | | | |

Samples (locality name, domiciliary unit identification). R: Remot; Wild: wild ecotope); Lat: Latadas; Clop: Cipriano Lopes; Quix: Quixabinha; Jen: Jenipapeiro; JDua: João Duarte.

abinha DU 5 was the only sample of origin that had two individuals reclassified. Samples from Latadas DU 18c1, Jenipapeiro DU 15 and João Duarte ecotope wild 3 (JDuaWild3) received two individuals; Latadas DU 70 from REMOT received three individuals (Table VI).

## DISCUSSION

Previous studies conducted in the North-East region of Brazil have reported that *T. brasiliensis* is the most prevalent triatomine in domestic environments.[55,56]

This species can form large colonies and has high levels of natural *T. cruzi* infection.[56]

In our study, the number of alleles per locus (two to 14) is lower than those observed by other authors studying the same species in north-east Brazil.[18,20] The population with the highest average number of alleles per locus (4.2) was of sylvatic origin (João Duarte locality, JDuaWild1), corroborating previous studies in the State of Ceará.[18] Almeida et al.[20] observed that the sylvatic populations they studied had higher average NA compared to peri-

TABLE IV

Values of observed heterozygosity (OH), expected heterozygosity (EH), and Hardy-Weinberg equilibrium (HW)
for each locus in *Triatoma brasiliensis* samples collected from Jaguaruana, Ceará, Brazil

| Sample\Locus | | Tb728 | Tb830 | Tb860 | Tb7180 | Tb8124 | HW p-value |
|---|---|---|---|---|---|---|---|
| Lat18c1 | OH | 0.30 | 0.40 | 0.00 | 0.11 | 0.00* | 0.0026* |
| | EH | 0.27 | 0.44 | 0.00 | 0.31 | 0.44* | |
| Lat23 | OH | 0.40 | 0.30* | 0.00 | 0.70 | 0.00* | 0.0477* |
| | EH | 0.34 | 0.58* | 0.00 | 0.62 | 0.19* | |
| Lat3c1 | OH | 0.33 | 0.80 | 0.20 | 0.60 | 0.30* | 0.0086* |
| | EH | 0.42 | 0.54 | 0.19 | 0.78 | 0.71* | |
| Lat11 | OH | 0.70 | 0.70 | 0.10 | 0.60 | 0.70* | 0.4404 |
| | EH | 0.72 | 0.57 | 0.10 | 0.64 | 0.79* | |
| Lat13 | OH | 0.30 | 0.40* | 0.40 | 0.20* | 0.10* | 0.0001* |
| | EH | 0.28 | 0.66* | 0.34 | 0.72* | 0.59* | |
| Lat14c1 | OH | 0.67 | 0.44 | 0.22 | 0.44* | 0.33* | 0.0005* |
| | EH | 0.73 | 0.52 | 0.37 | 0.79* | 0.75* | |
| CLop17 | OH | 0.43 | 0.71 | 0.57 | 0.14* | 0.43 | 0.0314* |
| | EH | 0.38 | 0.62 | 0.44 | 0.74* | 0.58 | |
| CLop33c1p1 | OH | 0.50 | 0.00* | 0.70 | 0.50* | 0.00* | 0.0016* |
| | EH | 0.39 | 0.34* | 0.48 | 0.77* | 0.19* | |
| CLop33c1p2 | OH | 0.40 | 0.10* | 0.30 | 0.70 | 0.20* | 0.0000* |
| | EH | 0.44 | 0.53* | 0.27 | 0.76 | 0.51* | |
| CLop15c2 | OH | 0.80 | 0.40 | 0.10 | 0.40 | 0.20 | 0.0128* |
| | EH | 0.63 | 0.34 | 0.10 | 0.72 | 0.34 | |
| CLop27 | OH | 0.50 | 0.40 | 0.50 | 1.00 | 0.20* | 0.1100 |
| | EH | 0.64 | 0.55 | 0.39 | 0.81 | 0.63* | |
| CLop23p1 | OH | 0.25 | 0.38* | 0.25 | 0.63 | 0.00* | 0.0008* |
| | EH | 0.23 | 0.69* | 0.23 | 0.78 | 0.52* | |
| CLop23p2 | OH | 0.10 | 0.70 | 0.00 | 0.40* | 0.20* | 0.0015* |
| | EH | 0.10 | 0.74 | 0.00 | 0.71* | 0.61* | |
| Quix5 | OH | 0.25 | 0.50 | 0.00 | 1.00 | 0.80 | 0.3424 |
| | EH | 0.25 | 0.68 | 0.00 | 0.71 | 0.82 | |
| Jen6 | OH | 0.57 | 0.00* | 0.57 | 0.14 | 0.00* | 0.0006* |
| | EH | 0.44 | 0.53* | 0.53 | 0.27 | 0.48* | |
| Jen1 | OH | 0.50 | 0.67 | 1.00 | 0.83 | 0.00* | 0.4950 |
| | EH | 0.41 | 0.53 | 0.55 | 0.80 | 0.48* | |
| Jen6c1 | OH | 0.60 | 0.00* | 0.60 | 0.80 | 0.00 | 0.2788 |
| | EH | 0.56 | 0.53* | 0.47 | 0.78 | 0.00 | |
| Jen15 | OH | 0.67 | 0.56 | 0.33 | 0.56 | 0.11* | 0.0400* |
| | EH | 0.58 | 0.57 | 0.42 | 0.76 | 0.50* | |
| JDuaWild1 | OH | 0.73 | 0.64* | 0.45 | 0.55 | 0.18* | 0.0022* |
| | EH | 0.77 | 0.68* | 0.39 | 0.77 | 0.63* | |
| JDuaWild2 | OH | 1.00 | 0.38 | 0.00 | 0.88* | 0.13* | 0.0953 |
| | EH | 0.71 | 0.53 | 0.00 | 0.88* | 0.63* | |
| JDuaWild3 | OH | 0.60 | 0.75 | 0.25 | 0.60 | 0.00 | 0.1477 |
| | EH | 0.64 | 0.54 | 0.25 | 0.82 | 0.43 | |
| JDuaWild4 | OH | 0.67 | 0.40 | 0.33 | 0.40* | 0.17* | 0.0003* |
| | EH | 0.67 | 0.53 | 0.32 | 0.82* | 0.80* | |
| CLopWild | OH | 0.43 | 0.43 | 0.71 | 0.71 | 0.00* | 0.1075 |
| | EH | 0.38 | 0.36 | 0.49 | 0.67 | 0.66* | |
| JDuaWild5 | OH | 0.33* | 0.56 | 0.14 | 0.44* | 0.22* | 0.0000* |
| | EH | 0.70* | 0.57 | 0.14 | 0.87* | 0.63* | |
| LatR25 | OH | 0.33 | 0.56 | 0.00 | 0.67 | 0.00* | 0.0001* |
| | EH | 0.45 | 0.66 | 0.23 | 0.81 | 0.68* | |
| LatR70 | OH | 0.40 | 0.40 | 0.50 | 0.60 | 0.00* | 0.0029* |
| | EH | 0.35 | 0.56 | 0.48 | 0.76 | 0.53* | |
| CLopR26 | OH | 0.40 | 0.33 | 0.33 | 1.00* | 0.10* | 0.1851 |
| | EH | 0.34 | 0.29 | 0.30 | 0.66* | 0.50* | |
| QuixR27 | OH | 0.20 | 0.40 | 0.50 | 0.50* | 0.10* | 0.0000* |
| | EH | 0.35 | 0.53 | 0.39 | 0.81* | 0.65* | |
| CLopR69 | OH | 0.40 | 0.20* | 0.00 | 0.50* | 0.20* | 0.0000* |
| | EH | 0.48 | 0.48* | 0.00 | 0.86* | 0.61* | |
| Average | OH | 0.47 | 0.41* | 0.32 | 0.56* | 0.16* | |
| | EH | 0.51 | 0.61* | 0.31 | 0.81* | 0.65* | |

*p ≤ 0.05. Samples (locality name, domiciliary unit identification). R: Remot; Wild: wild ecotope; Lat: Latadas; Clop: Cipriano Lopes;
Quix: Quixabinha; Jen: Jenipapeiro; JDua: João Duarte.

TABLE V

Analysis of molecular variance (AMOVA) and the fixation index for *Triatoma brasiliensis* collected
from the municipality of Jaguaruana in the State of Ceará, Brazil

| Variation source | Variation's components | Variation's percentage | Fixation index |
|---|---|---|---|
| Between populations | 0.14(Va) | 10.24 | 0.10* ($F_{st}$) |
| Between individuals within populations | 0.31 (Vb) | 22.57 | 0.25* ($F_{is}$) |
| Between individuals | 0.94 (Vc) | 67.19 | 0.33* ($F_{it}$) |

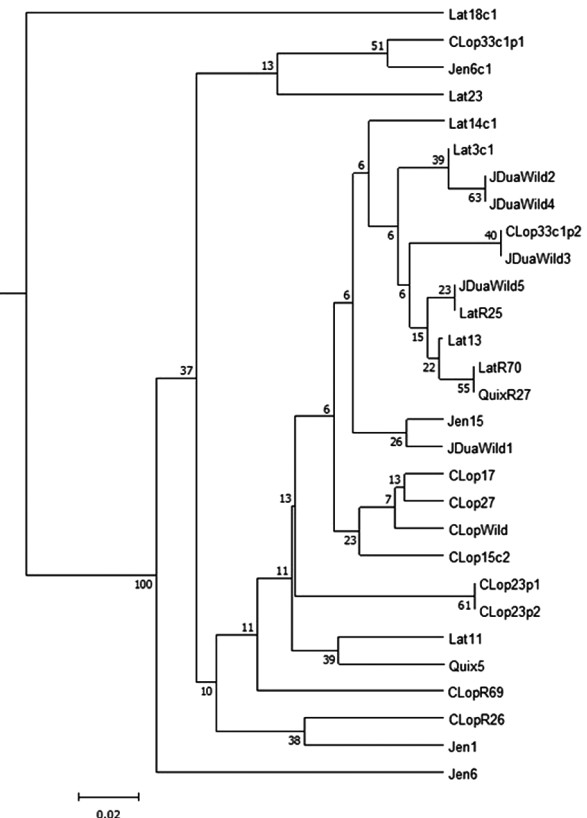

Fig. 3: dendogram Neighbour joing of Fst pairwise of *Triatoma brasiliensis* from Jaguaruana, Ceará. Samples (locality name, domiciliary unit identification). R: Remot; Wild: wild ecotope; Lat: Latadas; Clop: Cipriano Lopes; Quix: Quixabinha; Jen: Jenipapeiro; JDua: João Duarte.

domiciliary populations. In our study, the *T. brasiliensis* population from the Latadas locality (peridomicile, DU18c1) exhibited the lowest average NA and AR, along with the highest $F_{is}$ value (0.44), suggesting a heterozygosity deficit, the presence of null alleles, or population substructure. The heterozygosity deficit may result from persistent infestations by individuals that survived insecticide spraying, leading to increased inbreeding,[57] and/or from mating among individuals restricted to certain sylvatic habitats. The fixation indices further support this, as they showed significant values (p ≤ 0.05), indicating inbreeding, which may reflect both mating among related individuals and the presence of subpopulation structure.

In Jaguaruana, the most frequent sylvatic ecotope of *T. brasiliensis* is the cactus *Pilosocereus gounellei*, which is distributed in discontinuous clumps. This ecological context is contrary to that observed in Tauá, which differs from regions where triatomines are associated with granite outcrops (Fig. 1) and there is evidence of wide dispersal without cluster formation, characterising panmictic populations.[18]

The distribution pattern of triatomines in sylvatic environments certainly influences the reinfestation process in the anthropic and/or disturbed natural environments (*i.e.*, the intra- and peridomiciles). In Tauá, the area studied by Bezerra et al.,[18] the population density of triatomines within DUs recovers completely one year after spraying with residual insecticide. In contrast, in the municipality of Tamboril, also in the State of Ceará, and with a landscape similar to that of Jaguaruana, triatomine infestation remained low compared to original data over the same period.[58] These findings reinforce the ability and sensitivity of microsatellite markers for investigating the population dynamics of triatomines.

The NJ dendrogram suggests that the two peridomiciliary annexes of Cristiano Lopes' DU 23 (CLop23p1 and CLop23p2) had the same source of infestation or that one colonisation gave rise to another. A different situation occurred in the peridomiciliary annexes of DU 33 in the same locality (CLop33p1 and CLop33p2), where the sources of infestation probably were different. Although the branches did not show statistical support, the dendrogram suggests that the sylvatic focus found in Cristiano Lopes may have been responsible for the invasion of *T. brasiliensis* in DUs 17, 27, and 15c2 of the same locality (CLop17, CLop27, and CLop15c2). The Bayesian STRUCTURE analysis corroborated the dendrogram observations, both approaches provided complementary insights into population differentiation and gene flow, revealing multiple genetic clusters and varying degrees of admixture among the triatomine populations that we studied. The Lat18C1 population exhibited a homogeneous genetic composition, consistent with the differentiation observed in the NJ tree. The JDuaWild and Lat populations shared significant proportions of genetic ancestry, supporting their genetic proximity as inferred from the NJ analysis. In contrast, the CLop and Jen populations displayed a high degree of genetic admixture, suggesting a history of gene flow between these lineages. Additionally, some CLop populations (CLop23p1 and CLop23p2) exhibited similar genetic profiles, indicating a close and possibly recent relationship (Fig. 4).

TABLE VI

Geographic distances between the sampling locations in kilometres (above the diagonal), pairwise $F_{st}$ values (below the diagonal), and $F_{is}$ values (on the diagonal) for *Triatoma brasiliensis* collected from the municipality of Jaguaruana in the State of Ceará, Brazil

| Sample | Lat18c1 | Lat23 | Lat3c1 | Lat11 | Lat13 | Lat14c1 | CLop17 | CLop33c1p1 | CLop33c1p2 | CLop15c2 | CLop27 | CLop23p1 | CLop23p2 | Quix5 | Jen6 | Jen1 | Jen6c1 | Jen15 | JDuaWild1 | JDuaWild2 | JDuaWild3 | JDuaWild4 | CLopWild | JDuaWild5 | LatR25 | LatR70 | CLopR26 | QuixR27 | CLopR69 |
|---|---|---|---|---|---|---|---|---|---|---|---|---|---|---|---|---|---|---|---|---|---|---|---|---|---|---|---|---|---|
| Lat18c1 | 0.44 | 0.43 | 6.37 | 0.25 | 0.17 | 0.18 | 1.90 | 0.56 | 0.56 | 1.90 | 0.74 | 1.07 | 1.05 | 10.35 | 6.80 | 7.34 | 6.80 | 6.68 | 2.27 | 2.29 | 2.25 | 2.23 | 1.95 | 2.09 | 1.14 | 2.37 | 10.52 | 2.37 | 1.14 |
| Lat23 | 0.45* | 0.20* | 5.95 | 0.68 | 0.59 | 0.58 | 2.02 | 0.77 | 0.77 | 2.18 | 0.85 | 1.11 | 1.10 | 10.79 | 7.19 | 7.70 | 7.19 | 7.08 | 2.70 | 2.72 | 2.69 | 2.66 | 2.07 | 2.52 | 1.55 | 2.65 | 10.95 | 2.65 | 1.55 |
| Lat3c1 | 0.31* | 0.14* | 0.15* | 6.60 | 6.54 | 6.47 | 7.38 | 6.63 | 6.62 | 7.90 | 6.58 | 6.62 | 6.63 | 16.57 | 13.10 | 13.53 | 13.10 | 13.00 | 8.52 | 8.53 | 8.49 | 8.45 | 7.42 | 8.33 | 7.50 | 8.36 | 16.75 | 8.36 | 7.50 |
| Lat11 | 0.26* | 0.19* | 0.08* | 0.01* | 0.15 | 0.15 | 1.90 | 0.60 | 0.60 | 1.81 | 0.82 | 1.15 | 1.14 | 10.10 | 6.59 | 7.15 | 6.59 | 6.47 | 2.02 | 2.04 | 2.00 | 1.98 | 1.95 | 1.83 | 0.92 | 2.26 | 10.27 | 2.26 | 0.92 |
| Lat13 | 0.21* | 0.15* | 0.03 | 0.07* | 0.47 | 0.20 | 1.80 | 0.47 | 0.47 | 1.75 | 0.68 | 1.02 | 1.00 | 10.20 | 6.62 | 7.16 | 6.62 | 6.51 | 2.12 | 2.15 | 2.11 | 2.10 | 1.85 | 1.94 | 0.97 | 2.22 | 10.36 | 2.22 | 0.97 |
| Lat14c1 | 0.24* | 0.19* | 0.06 | 0.06* | 0.09 | 0.35 | 2.00 | 0.67 | 0.67 | 1.94 | 0.87 | 1.21 | 1.19 | 10.22 | 6.73 | 7.29 | 6.73 | 6.61 | 2.12 | 2.14 | 2.11 | 2.08 | 2.05 | 1.94 | 1.07 | 2.39 | 10.38 | 2.39 | 1.07 |
| CLop17 | 0.25* | 0.17* | 0.09* | 0.17* | 0.02 | 0.08 | 0.18* | 1.34 | 1.34 | 0.78 | 1.18 | 0.92 | 0.93 | 10.21 | 5.87 | 6.19 | 5.87 | 5.82 | 2.86 | 2.91 | 2.88 | 2.91 | 0.05 | 2.75 | 1.64 | 1.08 | 10.34 | 1.08 | 1.64 |
| CLop33c1p1 | 0.30* | 0.10* | 0.17* | 0.10* | 0.09* | 0.14* | 0.05 | 0.23* | 0.00 | 1.41 | 0.23 | 0.56 | 0.54 | 10.27 | 6.48 | 6.96 | 6.48 | 6.38 | 2.30 | 2.33 | 2.29 | 2.29 | 1.39 | 2.13 | 0.98 | 1.89 | 10.43 | 1.89 | 0.98 |
| CLop33c1p2 | 0.31* | 0.02 | 0.04 | 0.08* | 0.05 | 0.06 | 0.06 | 0.05 | 0.33 | 1.41 | 0.22 | 0.55 | 0.54 | 10.27 | 6.48 | 6.96 | 6.48 | 6.38 | 2.30 | 2.33 | 2.30 | 2.29 | 1.39 | 2.13 | 0.98 | 1.89 | 10.43 | 1.89 | 0.98 |
| CLop15c2 | 0.29* | 0.16* | 0.13* | 0.08* | 0.07 | 0.09* | 0.04 | 0.09* | 0.09* | 0.11* | 1.37 | 1.28 | 1.28 | 9.44 | 5.23 | 5.63 | 5.23 | 5.16 | 2.19 | 2.24 | 2.21 | 2.25 | 0.79 | 2.11 | 1.19 | 0.48 | 9.57 | 0.48 | 1.19 |
| CLop27 | 0.19* | 0.14* | 0.08* | 0.12* | 0.05 | 0.03 | 0.03 | 0.08* | 0.06 | 0.04 | 0.15 | 0.34 | 0.32 | 10.41 | 6.53 | 6.98 | 6.53 | 6.43 | 2.48 | 2.52 | 2.48 | 2.48 | 1.24 | 2.32 | 1.14 | 1.85 | 10.56 | 1.85 | 1.14 |
| CLop23p1 | 0.33* | 0.16* | 0.09* | 0.10* | 0.03 | 0.14* | 0.04 | 0.12* | 0.10* | 0.14* | 0.08 | 0.36 | 0.02 | 10.53 | 6.50 | 6.91 | 6.50 | 6.42 | 2.71 | 2.75 | 2.71 | 2.72 | 0.97 | 2.56 | 1.35 | 1.74 | 10.67 | 1.74 | 1.35 |
| CLop23p2 | 0.32* | 0.18* | 0.08 | 0.04 | 0.02 | 0.17* | 0.05 | 0.17* | 0.12* | 0.13* | 0.11* | -0.02 | 0.36 | 10.51 | 6.50 | 6.91 | 6.50 | 6.41 | 2.69 | 2.73 | 2.69 | 2.70 | 0.98 | 2.54 | 1.34 | 1.74 | 10.66 | 1.74 | 1.34 |
| Quix5 | 0.28* | 0.27* | 0.07 | 0.19* | 0.07 | 0.12* | 0.09 | 0.24* | 0.13* | 0.19* | 0.09 | 0.13* | 0.08 | -0.22* | 5.28 | 5.91 | 5.28 | 5.15 | 8.10 | 8.08 | 8.12 | 8.15 | 10.21 | 8.28 | 9.30 | 9.16 | 0.26 | 9.16 | 9.30 |
| Jen6 | 0.27* | 0.31* | 0.23* | 0.19* | 0.12 | 0.15* | 0.08 | 0.10 | 0.21* | 0.18* | 0.12* | 0.12 | 0.18* | 0.25* | 0.45 | 1.01 | 0.00 | 0.25 | 5.04 | 5.06 | 5.08 | 5.15 | 5.85 | 5.19 | 5.67 | 4.81 | 5.30 | 4.81 | 5.67 |
| Jen1 | 0.30* | 0.33* | 0.11* | 0.19* | 0.08 | 0.16* | 0.15* | 0.24* | 0.18* | 0.26* | 0.14* | 0.16* | 0.17* | 0.14* | 0.24* | -0.09* | 1.02 | 1.26 | 5.78 | 5.80 | 5.82 | 5.89 | 6.16 | 5.91 | 6.23 | 5.18 | 5.89 | 5.18 | 6.23 |
| Jen6c1 | 0.44* | 0.13* | 0.19* | 0.09* | 0.13 | 0.15* | 0.12 | 0.04 | 0.11 | 0.14* | 0.12* | 0.09 | 0.17* | 0.28* | 0.12 | 0.24* | 0.16* | 0.24 | 5.04 | 5.06 | 5.08 | 5.15 | 5.85 | 5.19 | 5.67 | 4.81 | 5.30 | 4.81 | 5.67 |
| Jen15 | 0.33* | 0.19* | 0.03 | 0.06* | 0.02 | 0.09* | 0.08 | 0.14* | 0.09* | 0.12* | 0.08* | 0.09 | 0.10* | 0.12* | 0.18* | 0.24* | 0.09 | 0.23 | 4.87 | 4.89 | 4.92 | 4.98 | 5.80 | 5.03 | 5.55 | 4.75 | 5.18 | 4.75 | 5.55 |
| JDuaWild1 | 0.31* | 0.18* | 0.08* | 0.08* | 0.09* | 0.04 | 0.10* | 0.15* | 0.11* | 0.11* | 0.09* | 0.06 | 0.11* | 0.11* | 0.15* | 0.04 | 0.06 | 0.02 | 0.22 | 0.05 | 0.05 | 0.12 | 2.89 | 0.19 | 1.35 | 2.31 | 8.27 | 2.31 | 1.35 |
| JDuaWild2 | 0.34* | 0.18* | 0.00 | 0.05 | 0.08 | 0.03 | 0.09 | 0.20* | 0.08 | 0.10* | 0.05 | 0.12* | 0.12* | 0.12* | 0.27* | 0.13* | 0.21* | 0.06 | 0.05 | 0.13* | 0.04 | 0.09 | 2.94 | 0.21 | 1.39 | 2.36 | 8.25 | 2.36 | 1.39 |
| JDuaWild3 | 0.29* | 0.10 | -0.05 | 0.06 | -0.02 | -0.03 | 0.02 | 0.07 | -0.05 | 0.05 | -0.02 | 0.07 | 0.08 | 0.10 | 0.17 | 0.15* | 0.11 | -0.01 | 0.03 | -0.03 | 0.11* | 0.07 | 2.91 | 0.17 | 1.36 | 2.35 | 8.29 | 2.35 | 1.36 |
| JDuaWild4 | 0.28* | 0.24* | -0.02 | 0.10* | 0.04 | 0.00 | 0.10 | 0.23* | 0.10 | 0.15* | 0.06 | 0.12 | 0.11* | 0.08 | 0.24* | 0.08 | 0.24* | 0.06 | 0.05 | -0.03 | -0.02 | 0.35 | 2.94 | 0.16 | 1.37 | 2.40 | 8.32 | 2.40 | 1.37 |
| CLopWild | 0.06 | 0.20* | 0.10* | 0.05 | 0.01 | 0.08 | 0.04 | 0.07 | 0.09* | 0.09* | 0.03 | 0.11 | 0.12* | 0.09 | 0.07 | 0.08 | 0.18* | 0.10 | 0.13* | 0.14* | 0.04 | 0.09 | 0.12* | 2.78 | 1.68 | 1.07 | 10.33 | 1.07 | 1.68 |
| JDuaWild5 | 0.20* | 0.16* | 0.01 | 0.04 | 0.01 | 0.01 | 0.07 | 0.12* | 0.04 | 0.07 | 0.02 | 0.09 | 0.10* | 0.06 | 0.18* | 0.11 | 0.14* | 0.01 | 0.04 | 0.00 | -0.05 | -0.02 | 0.03 | 0.43 | 1.20 | 2.27 | 8.46 | 2.27 | 1.20 |
| LatR25 | 0.20* | 0.10* | 0.01 | 0.13* | -0.01 | 0.07 | 0.04 | 0.08 | 0.03 | 0.09* | 0.04 | 0.02 | 0.04 | 0.03 | 0.15* | 0.05 | 0.10 | 0.02 | 0.04 | 0.04 | -0.02 | 0.02 | 0.02 | -0.01 | 0.46 | 1.55 | 9.46 | 1.55 | 0.00 |
| LatR70 | 0.20* | 0.17* | 0.03 | 0.16* | 0.01 | 0.09 | 0.07 | 0.09* | 0.04 | 0.14* | 0.05 | 0.10 | 0.11* | 0.10 | 0.16* | 0.07 | 0.15* | 0.02 | 0.12* | 0.09 | -0.04 | 0.05 | 0.02 | 0.00 | 0.00 | 0.30 | 9.28 | 0.00 | 1.55 |
| CLopR26 | 0.31* | 0.35* | 0.12* | 0.09* | 0.14* | 0.24* | 0.22* | 0.32* | 0.22* | 0.32* | 0.19* | 0.21* | 0.18* | 0.12* | 0.33* | 0.06 | 0.35* | 0.13* | 0.18* | 0.20* | 0.20* | 0.13* | 0.15* | 0.13* | 0.07 | 0.09* | -0.09* | 9.28 | 9.46 |
| QuixR27 | 0.17* | 0.22* | 0.02 | 0.09* | 0.00 | 0.08 | 0.04 | 0.15* | 0.08 | 0.13* | 0.05 | 0.08 | 0.06 | 0.04 | 0.17* | 0.02 | 0.21* | 0.04 | 0.11* | 0.06 | -0.02 | 0.00 | 0.02 | 0.00 | 0.00 | -0.02 | 0.07 | 0.39 | 1.55 |
| CLopR69 | 0.22* | 0.17* | 0.11* | 0.10* | 0.12* | 0.11* | 0.14* | 0.17* | 0.11* | 0.13* | 0.11* | 0.16* | 0.17* | 0.13 | 0.27* | 0.25* | 0.22* | 0.18* | 0.15* | 0.12* | 0.09 | 0.10 | 0.12* | 0.08 | 0.05 | 0.15* | 0.25* | 0.13* | 0.48 |

*p ≤ 0.05. Samples (locality name, domiciliary unit identification). R: Remot; Wild: wild ecotope; Lat: Latadas; Clop: Cipriano Lopes; Quix: Quixabinha; Jen: Jenipapeiro; JDua: João Duarte.

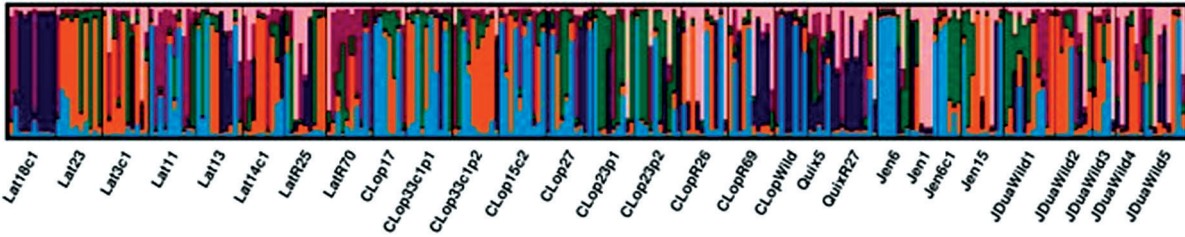

Fig. 4: bar chart representing genetic diversity for *Triatoma brasiliensis* from Jaguaruana, Ceará. Each bar represents an individual, and each colour represents one of the six clusters. Samples (locality name, domiciliary unit identification). R: Remot; Wild: wild ecotope; Lat: Latadas; Clop: Cipriano Lopes; Quix: Quixabinha; Jen: Jenipapeiro; JDua: João Duarte.

Susceptibility tests to pyrethroid insecticides show that the samples analysed here are susceptible to these insecticides (unpublished results obtained by REMOT), indicating that the persistence of triatomine infestation is not due to insecticide resistance. Our results emphasise the complexity of *T. brasiliensis* control, and highlight the difficulties and possible operational shortcomings, especially in the peridomiciliary environment. This fact is understandable given the numerous hiding places that are inaccessible to insecticide spraying both within the home and in their peridomiciliary annexes, even considering the residual activity of the insecticide indoors.[13,59,60]

In conclusion, our study provides key insights into the genetic structure and population dynamics of *T. brasiliensis* in Jaguaruana. The observed complexity, with anthropogenic environments colonised from diverse sources, emphasises the challenges faced in vector control. This necessitates tailored strategies that consider regional variations and the adaptability of *T. brasiliensis*. Effective surveillance and control planning must address not only existing infestations but also the intricate processes influencing vector dynamics. By enhancing our understanding of these complexities, we pave the way for more targeted and sustainable CD control efforts in our study region.

### ACKNOWLEDGEMENTS

To the endemic disease control agents of Jaguaruana, Decentralised Health Area - Russas (CE), particularly Márcia Lúcia de Oliveira Gomes (Coordinator) and Francisca Samya Silva de Freitas. We also thank the Vector Control Centre of the Health Department of the State of Ceará for their collaboration. We are thankful to the technicians of the DNA Sequencing Platform at IRR/Fiocruz Minas.

### AUTHORS' CONTRIBUTION

All authors contributed extensively to the work presented in this paper. Experimental design: CJB, LD, CMB, LORS; insect collection: LORS, CMB; experiment development: CJB and FCF; data analysis: CJB, LORS, CMB, LD, FCF and JH; drafting and revising the article: LORS, CJB, CMB and JH with significant contributions from the other authors. All authors read, approved the final version and consent to the publication of the manuscript. The authors declare no conflict of interest.

### DATA AVAILABILITY

The datasets used and/or analysed during the current study are available from the corresponding author on reasonable request.

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

# OPEN PEER REVIEW

Memórias do IOC thanks the anonymous reviewers for their contribution to the peer review of this work.

## FIRST REVIEW ROUND

REVIEWERS' COMMENTS

### REVIEWER #1

The manuscript presents an original and relevant study on the genetic structure of Triatoma brasiliensis populations in the municipality of Jaguaruana, located in the semi-arid region of Ceará, Brazil. The methodological approach is sound, combining representative sampling in domestic, peridomestic, and sylvatic environments with robust population genetic analyses using microsatellite markers.

The topic addressed is of great public health importance, particularly in regions endemic for Chagas disease, as it highlights the complexity of infestation and reinfestation dynamics in areas subjected to varying environmental pressures. The identification of significant genetic variability among geographically proximate populations and the absence of correlation with physical distance underscore the multifactorial nature of T. brasiliensis colonization, with direct implications for the design of surveillance and vector control strategies.

The population structure analyses—including AMOVA, dendrograms, and Bayesian inferences (STRUCTURE)—were conducted appropriately. The finding of multiple infestation sources and gene flow between ecotopes suggests that chemical control alone is insufficient to contain this species, especially in peridomestic areas.

Genetic data point to deviations from Hardy-Weinberg equilibrium in several samples, indicating an excess of homozygotes and probable inbreeding. These aspects may be related to the local persistence of colonies and recolonization by surviving individuals following control efforts.

However, some aspects deserve attention to enhance the clarity and impact of the article:

The language and writing style need refinement. While the text is understandable, the English contains constructions that would benefit from revision. It is strongly recommended that the manuscript be reviewed by a native English speaker to ensure grammatical accuracy and fluency.

There are references to "Supplement 1" and "Supplement 2," which appear to be essential for verifying allele sizes and null allele frequencies. These documents were not included in the submitted files. Their absence partially compromises the full assessment of the results' consistency.

The discussion of insecticide susceptibility is based on "unpublished data." It would be advisable to either cite the source or clarify that this is a preliminary observation. If these data are in preparation for publication, this should be explicitly stated.

The conclusion could be more concise and actionable. While the manuscript emphasizes the need for locally adapted strategies, it lacks concrete suggestions on how the genetic findings could be used to inform vector control programs.

A brief reflection at the end of the discussion regarding the study's limitations—such as the limited number of polymorphic loci, the lack of broader temporal comparisons, or challenges in extrapolation—would enrich the critical analysis of the work.

Additionally, the quality of all figures is very poor, making it difficult to properly evaluate the visual data. The images should be revised and replaced with higher-resolution versions that meet the journal's standards for publication.

Despite these remarks, the manuscript is scientifically sound and provides a meaningful contribution to the field of medical entomology, particularly regarding Chagas disease surveillance and control in rural Brazil. It is a well-designed study that integrates ecological and genetic data with competence.

I recommend acceptance of the manuscript pending minor revisions, particularly concerning language polishing, inclusion of supplementary data, improvement of figure quality, and minor adjustments in the presentation of results and conclusions. I would like to review the revised version before final acceptance.

### REVIEWER #2

Dear Editor,

Please receive my comments regarding the manuscript "The Complexity of the Population Dynamics of Triatoma brasiliensis in Rural Northeast, Brazil, Indicated by Genetic Characterization". Overall, the manuscript presents important results that contribute to the understanding of Triatoma brasiliensis, a vector of Chagas disease, particularly concerning its genetic characteristics in northeastern Brazil. However, there are several important issues that the authors must address and correct before the manuscript can be fully accepted for publication in MIOC.

Major issues

In the cover letter, the corresponding author highlights an important point that is omitted from the introduction and should be incorporated: the study area has a different landscape compared to previously studied ones. This aspect should also be discussed in the appropriate section.

Although the authors performed several population genetic analyses, the results are poorly discussed. Is there significant population structure or not? Does it vary according to environmental differences with other regions?

Many inferences are based on the NJ tree. However, several of the discussed groups are not statistically supported. Only four significant clusters (with bootstrap support higher than 50%) are observed: CLop33c1p1 + Jen6c1, JDuaWild2 + JDuaWild4, LatR76 + QuixR27, and CLop23p1 + CLop23p2. Additionally, you should specify that the tree was arbitrarily rooted, as no outgroup was used.

The description of the Structure clusters should be moved from the Discussion to the Results section. Which statistic was used in the Puechmaille method to select the K value? And which was used in Structure Selector? Furthermore, it is difficult to identify the patterns you mention, given the high levels of admixture and the heterogeneity of individuals within DU.

The AMOVA analysis could be improved by incorporating the hierarchical level of location (i.e., location, DU within location, and individuals within DU). You could also consider comparing peridomestic and sylvatic populations.

I disagree with the idea that departures from Hardy-Weinberg equilibrium are solely due to inbreeding. If that were the case, such a pattern should be observed across all loci. However, locus Tb8124 shows departures in nearly all populations, strongly indicating the presence of null alleles. In some populations (e.g., Lat13, CLop33c1p1, JDuaWild5, CLopR69), inbreeding or substructuring may be present, as suggested by departures in three loci.

Minor issues

L50-1: ...but vertical transmission has contributed substantially to the spread of this disease to non-endemic areas.

L54: This implies that approximately 83% of infected individuals live in Brazil. Is that accurate?

L57: Replace "four" with "four species".

L60: Change "Brasilian" to "Brazilian".

L60-1: Please rephrase; the sentence appears out of context. Consider clarifying how it connects with the previous content.

L62: Do you mean "domicile"?

L68: Delete "with infection".

L84-6: Please rephrase.

L90: Citing 13 references is excessive. Please select 4–5 representative examples.

L120: Figure 2 could be moved to supplementary material, as the manuscript do not discuss genetic differences between ecotopes.

L 135-6: "it also analyzed frozen insects", rewrite.

L171: Which statistical test was used to assess Hardy-Weinberg equilibrium?

L193: Rather than "verify," it would be more accurate to say you "analyzed" or "studied" genetic structure.

L 205: Please correct the grammar in this sentence.

L220-2: Please explain the NAE method more clearly. Are these values global FSTs?

L225-6: Inbreeding and substructuring are distinct phenomena. Additionally, the presence of null alleles should be considered.

L241: You could consider performing a Mantel test within localities to assess isolation by distance at a finer scale.

L263: The migration test does not appear to contribute any significant results to the main discussion. Migration between localities is not addressed elsewhere.

L513: Table II could be moved to the supplementary material.

L534: Table VI could be moved to the supplementary material or omitted. $F_{ST}$ values are already provided in Supplementary Table 1, and geographic distances are not relevant as the Mantel test was non-significant.

## AUTHORS' RESPONSE TO THE REVIEWERS

Dear Reviewers,

We sincerely thank you for your insightful comments and constructive suggestions, which have significantly improved the quality of our manuscript. The majority of the recommendations were incorporated, and the corresponding modifications are highlighted in the revised version, with the exception of the extensive edits resulting from the English language revision. Detailed, point-by-point responses to each of your comments are provided below.

We truly appreciate the time and effort dedicated to reviewing our work.

REVIEWER COMMENTS:

Reviewer: 1

The manuscript presents an original and relevant study on the genetic structure of Triatoma brasiliensis populations in the municipality of Jaguaruana, located in the semi-arid region of Ceará, Brazil. The methodological approach is sound, combining representative sampling in domestic, peridomestic, and sylvatic environments with robust population genetic analyses using microsatellite markers.

The topic addressed is of great public health importance, particularly in regions endemic for Chagas disease, as it highlights the complexity of infestation and reinfestation dynamics in areas subjected to varying environmental pressures. The identification of significant genetic variability among geographically proximate populations and the absence of correlation with physical distance underscore the multifactorial nature of T. brasiliensis colonization, with direct implications for the design of surveillance and vector control strategies.

The population structure analyses—including AMOVA, dendrograms, and Bayesian inferences (STRUCTURE)—were conducted appropriately. The finding of multiple infestation sources and gene flow between ecotopes suggests that chemical control alone is insufficient to contain this species, especially in peridomestic areas.

Genetic data point to deviations from Hardy-Weinberg equilibrium in several samples, indicating an excess of homozygotes and probable inbreeding. These aspects may be related to the local persistence of colonies and recolonization by surviving individuals following control efforts.

However, some aspects deserve attention to enhance the clarity and impact of the article:

The language and writing style need refinement. While the text is understandable, the English contains constructions that would benefit from revision. It is strongly recommended that the manuscript be reviewed by a native English speaker to ensure grammatical accuracy and fluency.

The manuscript was revised by a native English speaker.

There are references to "Supplement 1" and "Supplement 2," which appear to be essential for verifying allele sizes and null allele frequencies. These documents were not included in the submitted files. Their absence partially compromises the full assessment of the results' consistency.

The supplementary materials were previously provided as appendices.

The discussion of insecticide susceptibility is based on "unpublished data." It would be advisable to either cite the source or clarify that this is a preliminary observation. If these data are in preparation for publication, this should be explicitly stated.

This data was generated by REMOT, and this information has now been incorporated into the manuscript.

The conclusion could be more concise and actionable. While the manuscript emphasizes the need for locally adapted strategies, it lacks concrete suggestions on how the genetic findings could be used to inform vector control programs.

A brief reflection at the end of the discussion regarding the study's limitations—such as the limited number of polymorphic loci, the lack of broader temporal comparisons, or challenges in extrapolation—would enrich the critical analysis of the work.

The discussion is indeed relevant; nevertheless, our results indicate that, despite the limited number of markers available, it was still possible to reveal distinct dispersion patterns across different ecological contexts. In our previous study in the municipality of Tauá, the sample exhibited a panmictic pattern, which contrasts with the population subdivision detected in Jaguaruana when using the same molecular markers.

Additionally, the quality of all figures is very poor, making it difficult to properly evaluate the visual data. The images should be revised and replaced with higher-resolution versions that meet the journal's standards for publication.

The figures have also been provided as an appendix, in the resolution required by the journal.

Despite these remarks, the manuscript is scientifically sound and provides a meaningful contribution to the field of medical entomology, particularly regarding Chagas disease surveillance and control in rural Brazil. It is a well-designed study that integrates ecological and genetic data with competence.

I recommend acceptance of the manuscript pending minor revisions, particularly concerning language polishing, inclusion of supplementary data, improvement of figure quality, and minor adjustments in the presentation of results and conclusions. I would like to review the revised version before final acceptance.

Reviewer: 2

Dear Editor,

Please receive my comments regarding the manuscript "The Complexity of the Population Dynamics of Triatoma brasiliensis in Rural Northeast, Brazil, Indicated by Genetic Characterization". Overall, the manuscript presents important results that contribute to the understanding of Triatoma brasiliensis, a vector of Chagas disease, particularly concerning its genetic characteristics in northeastern Brazil. However, there are several important issues that the authors must address and correct before the manuscript can be fully accepted for publication in MIOC.

Major issues

In the cover letter, the corresponding author highlights an important point that is omitted from the introduction and should be incorporated: the study area has a different landscape compared to previously studied ones. This aspect should also be discussed in the appropriate section.

Done.

Although the authors performed several population genetic analyses, the results are poorly discussed. Is there significant population structure or not? Does it vary according to environmental differences with other regions?

The discussion has been appropriately revised, providing greater clarity to the aspect under consideration.

Many inferences are based on the NJ tree. However, several of the discussed groups are not statistically supported. Only four significant clusters (with bootstrap support higher than 50%) are observed: CLop33c1p1 + Jen6c1, JDuaWild2 + JDuaWild4, LatR76 + QuixR27, and CLop23p1 + CLop23p2. Additionally, you should specify that the tree was arbitrarily rooted, as no outgroup was used.

The necessary clarifications have been included in the text.

The description of the Structure clusters should be moved from the Discussion to the Results section.

Done

Which statistic was used in the Puechmaille method to select the K value? And which was used in Structure Selector? Furthermore, it is difficult to identify the patterns you mention, given the high levels of admixture and the heterogeneity of individuals within DU.

Puchamaille's method uses the values of median of means, maximum of means, median of medians, and maximum of medians. Evanno's method, which uses only DeltaK values. Both methods are reliable, but each has its own particularities that must be evaluated for its data set. The StructureSelector program has both methods (Evanno and Puchamaille).

The AMOVA analysis could be improved by incorporating the hierarchical level of location (i.e., location, DU within location, and individuals within DU). You could also consider comparing peridomestic and sylvatic populations.

We conducted a hierarchical-level AMOVA analysis as suggested; nevertheless, the interpretation of the results is consistent with that already presented in the manuscript. Consequently, we opted not to include these additional analyses in the text.

I disagree with the idea that departures from Hardy-Weinberg equilibrium are solely due to inbreeding. If that were the case, such a pattern should be observed across all loci. However, locus Tb8124 shows departures in nearly all populations, strongly indicating the presence of null alleles. In some populations (e.g., Lat13, CLop33c1p1, JDuaWild5, CLopR69), inbreeding or substructuring may be present, as suggested by departures in three loci.

It is true that deviations from Hardy–Weinberg equilibrium may also result from null alleles. This possibility was investigated, and our results indicate that the null alleles present in the sample do not influence the outcomes of the analysis, as reported in the Results section.

Minor issues

L50-1: ...but vertical transmission has contributed substantially to the spread of this disease to non-endemic areas. Done.

L54: This implies that approximately 83% of infected individuals live in Brazil. Is that accurate?

The statement that approximately 83% of infected individuals live in Brazil is not supported by the references used in the manuscript. According to WHO (2018, 2022) and the 2nd Brazilian Consensus on Chagas Disease (2015), it is estimated that around 6 million people are infected worldwide, while Brazil accounts for between 1.9 and 4.6 million of these cases. This corresponds to approximately 32% to 77% of global infections.

L57: Replace "four" with "four species". Done.

L60: Change "Brasilian" to "Brazilian". Done

L60-1: Please rephrase; the sentence appears out of context. Consider clarifying how it connects with the previous content. Done

L62: Do you mean "domicile"?

The term "domiciliary" is correct as it complements the word "ecotope" in the context of the sentence. However, it can be changed if it improves clarity for the reader.

L68: Delete "with infection". Done

L84-6: Please rephrase. Done

L90: Citing 13 references is excessive. Please select 4–5 representative examples.

Thank you for your suggestion. While I understand the recommendation to limit the number of references, I believe that citing all 13 studies is important to fully support the statement, as it reflects the diversity and breadth of research conducted on gene flow, vector dispersal, and taxonomic assessment of vectors. Each of these studies contributes unique insights to the topic and including them provides a comprehensive overview.

L120: Figure 2 could be moved to supplementary material, as the manuscript do not discuss genetic differences between ecotopes.

The figure was actually not adequate. We changed the figure to demonstrate the differences between the two ecotopes.

L 135-6: "it also analyzed frozen insects", rewrite. Done

L171: Which statistical test was used to assess Hardy-Weinberg equilibrium?

The statistical test used by Arlequin to assess Hardy-Weinberg equilibrium is the exact Hardy-Weinberg test, which is based on probability values. Specifically, Arlequin implements the exact test developed by Guo and Thompson (1992), which evaluates whether the observed allele frequencies significantly differ from those expected under Hardy-Weinberg equilibrium. This test is particularly useful for small sample sizes and provides a reliable way to assess genetic equilibrium in populations.

L193: Rather than "verify," it would be more accurate to say you "analyzed" or "studied" genetic structure. Done

L 205: Please correct the grammar in this sentence. Done

L220-2: Please explain the NAE method more clearly. Are these values global FSTs? The ENA method is described in Chapuis and Estoup 2007. Basically, it uses the Fst estimate described by Weir (1996) for the entire set of microsatellites, as it allows greater accuracy in estimating these values in the presence of null alleles, and the correction is made by the method described by Cavalli-Sforza and Edwards (1967). Through the use of bootstrap, the lowest and highest values of the 95% confidence intervals are obtained, taking the quantiles of 2.5% and 97.5% respectively. The global Fst (both with null alleles and without null alleles) obtained must include both values for the method to be reliable.

L225-6: Inbreeding and substructuring are distinct phenomena. Additionally, the presence of null alleles should be considered.

The presence of null alleles is indeed an important consideration and has been discussed in the Discussion section.

L241: You could consider performing a Mantel test within localities to assess isolation by distance at a finer scale.

Fine-scale analyses, including Mantel tests within localities to assess isolation by distance, were performed; however, no significant relationships were detected.

L263: The migration test does not appear to contribute any significant results to the main discussion. Migration between localities is not addressed elsewhere.

The migration test results corroborate the other analyses, suggesting low migration among subpopulations. Accordingly, we deemed it important to include these results in the manuscript.

L513: Table II could be moved to the supplementary material.

L534: Table VI could be moved to the supplementary material or omitted. FST values are already provided in Supplementary Table 1, and geographic distances are not relevant as the Mantel test was non-significant.

Regarding the last comments, we understand your point of view; however, we ask to maintain the format traditionally used in our publications. We consider these tables to contain information that is frequently consulted, especially for studies on the same topic.

## SECOND REVIEW ROUND

### REVIEWERS' COMMENTS

#### REVIEWER #1

No other comments.

#### REVIEWER #2

Please receive my comments regarding the revised version of this manuscript. To better understand the revisions, in several cases I have copied my original question, the authors' answers, and my reply, because in many cases my proposals were not properly understood or addressed. I have also added some new comments.

Major issues

In the cover letter, the corresponding author highlights an important point that is omitted from the introduction and should be incorporated: the study area has a different landscape compared to previously studied ones. This aspect should also be discussed in the appropriate section.

Done.

Regarding this topic:

L 81-82. Change "within different natural ecotope of Jaguaruana to "with a different natural ecotope from Jaguaruana"

Many inferences are based on the NJ tree. However, several of the discussed groups are not statistically supported. Only four significant clusters (with bootstrap support higher than 50%) are observed: CLop33c1p1 + Jen6c1, JDuaWild2 + JDuaWild4, LatR76 + QuixR27, and CLop23p1 + CLop23p2. Additionally, you should specify that the tree was arbitrarily rooted, as no outgroup was used.

The necessary clarifications have been included in the text.

L192-193. Delete "this tree lacks statistical support". It is not correct. Some clusters are statistically supported while others are not and this should be informed in Results.

In addition, the authors insist on reporting and discussing the implications of clusters not statistically supported (Results L258–263, Discussion L314–317).

Which statistic was used in the Puechmaille method to select the K value? And which was used in Structure Selector? Furthermore, it is difficult to identify the patterns you mention, given the high levels of admixture and the heterogeneity of individuals within DU.

Puchamaille's method uses the values of median of means, maximum of means, median of medians, and maximum of medians. Evanno's method, which uses only DeltaK values. Both methods are reliable, but each has its own particularities that must be evaluated for its data set. The StructureSelector program has both methods (Evanno and Puchamaille).

The authors explained the method in general but did not answer my question: Which statistic(s) did you use? Is there a graphic or p-value that shows why you chose K = 6? If so, please provide it.

The AMOVA analysis could be improved by incorporating the hierarchical level of location (i.e., location, DU within location, and individuals within DU). You could also consider comparing peridomestic and sylvatic populations.

We conducted a hierarchical-level AMOVA analysis as suggested; nevertheless, the interpretation of the results is consistent with that already presented in the manuscript. Consequently, we opted not to include these additional analyses in the text.

It is unfortunate that the authors did not incorporate these results into the manuscript. You have a sampling structure completely suitable for this. AMOVA is a hierarchical test: if you only use the level "among populations" you already have the pairwise Fsts, so a global Fst adds almost no information.

I disagree with the idea that departures from Hardy-Weinberg equilibrium are solely due to inbreeding. If that were the case, such a pattern should be observed across all loci. However, locus Tb8124 shows departures in nearly all populations, strongly indicating the presence of null alleles. In some populations (e.g., Lat13, CLop33c1p1, JDuaWild5, CLopR69), inbreeding or substructuring may be present, as suggested by departures in three loci.

It is true that deviations from Hardy–Weinberg equilibrium may also result from null alleles. This possibility was investigated, and our results indicate that the null alleles present in the sample do not influence the outcomes of the analysis, as reported in the Results section

The authors are mistaken on this point. They tend to mix up Fst with Fis results. The ENA correction was applied to Fst estimates. But my concern was that they did not incorporate the possibility that null alleles explain departures from HWE within populations, which directly affects Fis. Null alleles typically generate an excess of homozygotes, which is indeed observed here. I could not access the null alleles file in Supplement 2.

In addition, please order the Results section logically. Currently:
• Lines 228–233: HW departures
• Lines 234–238: effects of null alleles on pairwise Fst
• Lines 239–242: AMOVA
• Lines 242–254: return to Fis (HW departures)
• Lines 243-254: return to Fst

Minor issues

L171: Which statistical test was used to assess Hardy-Weinberg equilibrium?

The statistical test used by Arlequin to assess Hardy-Weinberg equilibrium is the exact Hardy-Weinberg test, which is based on probability values. Specifically, Arlequin implements the exact test developed by Guo and Thompson (1992), which evaluates whether the observed allele frequencies significantly differ from those expected under Hardy-Weinberg equilibrium. This test is particularly useful for small sample sizes and provides a reliable way to assess genetic equilibrium in populations.

You should incorporate this information in the manuscript. A single line stating that you used the Guo and Thompson exact test is sufficient.

L193: Rather than "verify," it would be more accurate to say you "analyzed" or "studied" genetic structure.

Done

This correction was not made

L225-6: Inbreeding and substructuring are distinct phenomena. Additionally, the presence of null alleles should be considered.

The presence of null alleles is indeed an important consideration and has been discussed in the Discussion section.

This is incorrect. The presence of null alleles is not discussed in the Discussion section.

L263: The migration test does not appear to contribute any significant results to the main discussion. Migration between localities is not addressed elsewhere.

The migration test results corroborate the other analyses, suggesting low migration among subpopulations. Accordingly, we deemed it important to include these results in the manuscript.

Despite the insistence on keeping the migration results, not a single word is devoted to discussing them.

New comments:

L 85-88. Change the sentence "In the state of…" to "In the state of Paraíba, also in northeastern Brazil, Almeida et al. (19) used the mitochondrial cytb gene and, in contrast to the study described above, suggested that T. brasiliensis populations are genetically structured."

L194: As previously requested, please add the method used to test departures from HWE. You explained this in your response but did not incorporate it into the text.

L209. Delete "from"

L210. Change "Markov Chain and Monte Carlo" to "Monte Carlo Markov Chain"

L211. Ancestral mixture model is not correct. Change "correlated allele frequencies"

L235. This is not the only point regarding null alleles. The issue is that you declare substructuring in the Discussion, when null alleles could be generating the departures from HWE. This possibility should be properly addressed.

L288-289. Or null alleles

Figures and Tables

Figure 1. The idea is very good. Could you make it tighter? Panel B is unequally distributed within the black frame.

Figure 2 legend. Specify to which locality each number corresponds.

Table IV. Contains several errors (e.g., a p-value of 0.0314 in Clop17 is not marked as significant; a p-value of 0.0128 in Clop15c2 is also unmarked, and others).

Tables. Seven tables are excessive. Please (again) consider moving some to the supplementary material.

## AUTHORS' RESPONSE TO THE REVIEWERS

REVIEWER COMMENTS:

Reviewer: 2

Please receive my comments regarding the revised version of this manuscript. To be er understand the revisions, in several cases I have copied my original ques on, the authors' answers, and my reply, because in many cases my proposals were not properly understood or addressed. I have also added some new comments.

Major issues

In the cover le er, the corresponding author highlights an important point that is omi ed from the introduc on and should be incorporated: the study area has a different landscape compared to previously studied ones. This aspect should also be discussed in the appropriate sec on.

Done.

Regarding this topic:

L 81-82. Change "within different natural ecotope of Jaguaruana to "with a different natural ecotope from Jaguaruana" Done.

Many inferences are based on the NJ tree. However, several of the discussed groups are not sta s cally supported. Only four significant clusters (with bootstrap support higher than 50%) are observed: CLop33c1p1 + Jen6c1, JDuaWild2 + JDuaWild4, LatR76 + QuixR27, and CLop23p1 + CLop23p2. Addi onally, you should specify that the tree was arbitrarily rooted, as no outgroup was used.

The necessary clarifica ons have been included in the text.

L192-193. Delete "this tree lacks sta s cal support". It is not correct. Some clusters are sta s cally supported while others are not and this should be informed in Results. This has been corrected.

In addi on, the authors insist on repor ng and discussing the implica ons of clusters not sta s cally supported (Results L258–263, Discussion L314–317). "The result concerning João Duarte (JDuaWild5) and Latadas REMOT (LaR25) was excluded, and only groupings with bootstrap values greater than 50 were retained."

Which sta s c was used in the Puechmaille method to select the K value? And which was used in Structure Selector? Furthermore, it is difficult to iden fy the pa erns you men on, given the high levels of admixture and the heterogeneity of individuals within DU.

Puchamaille's method uses the values of median of means, maximum of means, median of medians, and maximum of medians. Evanno's method, which uses only DeltaK values. Both methods are reliable, but each has its own par culari es that must be evaluated for its data set. The StructureSelector program has both methods (Evanno and Puchamaille).

The authors explained the method in general but did not answer my ques on: Which sta s c(s) did you use? Is there a graphic or p-value that shows why you chose K = 6? If so, please provide it. The sta s cs used were added to the manuscript text, and the graphs were included as Supplement 3.

The AMOVA analysis could be improved by incorpora ng the hierarchical level of loca on (i.e., loca on, DU within loca on, and individuals within DU). You could also consider comparing peridomes c and sylva c popula ons.

We conducted a hierarchical-level AMOVA analysis as suggested; nevertheless, the interpreta on of the results is consistent with that already presented in the manuscript. Consequently, we opted not to include these addi onal analyses in the text.

It is unfortunate that the authors did not incorporate these results into the manuscript. You have a sampling structure completely suitable for this. AMOVA is a hierarchical test: if you only use the level "among popula ons" you already have the pairwise Fsts, so a global Fst adds almost no informa on. We appreciate the reviewer's sugges on. While hierarchical AMOVA is indeed informa ve, our study was designed to assess gene c structure primarily at the DU level, which is directly addressed by the AMOVA and pairwise Fst results already presented. Exploratory hierarchical AMOVA produced pa erns consistent with those reported, without altering the interpreta on. To maintain focus and avoid redundancy, we retained the current analy cal framework, which adequately addresses the study objec ves.

I disagree with the idea that departures from Hardy-Weinberg equilibrium are solely due to inbreeding. If that were the case, such a pa ern should be observed across all loci. However, locus Tb8124 shows departures in nearly all popula ons, strongly indica ng the presence of null alleles. In some popula ons (e.g., Lat13, CLop33c1p1, JDuaWild5, CLopR69), inbreeding or substructuring may be present, as suggested by departures in three loci.

It is true that devia ons from Hardy–Weinberg equilibrium may also result from null alleles. This possibility was inves gated, and our results indicate that the null alleles present in the sample do not influence the outcomes of the analysis, as reported in the Results sec on.

The authors are mistaken on this point. They tend to mix up Fst with Fis results. The ENA correc on was applied to Fst es mates. But my concern was that they did not incorporate the possibility that null alleles explain departures from HWE within popula ons, which directly affects Fis. Null alleles typically generate an excess of homozygotes, which is indeed observed here. I could not access the null alleles file in Supplement 2. Yes, it was indeed confusing. The text has been reorganized, and I hope it now aligns with your considera ons.

In addi on, please order the Results sec on logically. Currently:
• Lines 228–233: HW departures
• Lines 234–238: effects of null alleles on pairwise Fst
• Lines 239–242: AMOVA
• Lines 242–254: return to Fis (HW departures)
• Lines 243-254: return to Fst
The effects of null alleles were moved to the end of the AMOVA results sec on.

Minor issues
L171: Which sta s cal test was used to assess Hardy-Weinberg equilibrium?
The sta s cal test used by Arlequin to assess Hardy-Weinberg equilibrium is the exact Hardy Weinberg test, which is based on probability values. Specifically, Arlequin implements the exact test developed by Guo and Thompson (1992), which evaluates whether the observed allele frequencies significantly differ from those expected under Hardy-Weinberg equilibrium. This test is par cularly useful for small sample sizes and provides a reliable way to assess gene c equilibrium in popula ons.

You should incorporate this informa on in the manuscript. A single line sta ng that you used the Guo and Thompson exact test is sufficient. Done.
L193: Rather than "verify," it would be more accurate to say you "analyzed" or "studied" gene c structure. Done.
This correc on was not made Done.
L225-6: Inbreeding and substructuring are dis nct phenomena. Addi onally, the presence of null alleles should be considered.
The presence of null alleles is indeed an important considera on and has been discussed in the Discussion sec on.
This is incorrect. The presence of null alleles is not discussed in the Discussion sec on.
L263: The migra on test does not appear to contribute any significant results to the main discussion. Migra on between locali es is not addressed elsewhere.
The migra on test results corroborate the other analyses, sugges ng low migra on among subpopula ons. Accordingly, we deemed it important to include these results in the manuscript.
Despite the insistence on keeping the migra on results, not a single word is devoted to discussing them. Table VII was removed from the manuscript, and only the descrip on of the results remained in the text. Indeed, the migra on test was not included in the discussion; its result was added to support the evidence of low migra on among popula ons.

New comments:
L 85-88. Change the sentence "In the state of…" to "In the state of Paraíba, also in northeastern Brazil, Almeida et al. (19) used the mitochondrial cytb gene and, in contrast to the study described above, suggested that T. brasiliensis popula ons are gene cally structured." Done.
L194: As previously requested, please add the method used to test departures from HWE. You explained this in your response but did not incorporate it into the text. Done.
L209. Delete "from" Done.

L210. Change "Markov Chain and Monte Carlo" to "Monte Carlo Markov Chain" Done.

L211. Ancestral mixture model is not correct. Change "correlated allele frequencies" Done.

L235. This is not the only point regarding null alleles. The issue is that you declare substructuring in the Discussion, when null alleles could be genera ng the departures from HWE. This possibility should be properly addressed. Done.

L288-289. Or null alleles Done.

Figures and Tables

Figure 1. The idea is very good. Could you make it ghter? Panel B is unequally distributed within the black frame. Done.

Figure 2 legend. Specify to which locality each number corresponds. The locali es were already specified in the figure legend.

Table IV. Contains several errors (e.g., a p-value of 0.0314 in Clop17 is not marked as significant; a p-value of 0.0128 in Clop15c2 is also unmarked, and others). Significant p-values were only marked in places where they were not originally shown; however, for clarity, we have now added * to all values $\leq 0.05$.

Tables. Seven tables are excessive. Please (again) consider moving some to the supplementary material. We have removed Table VII, and we hope the revision is now suitable.

## THIRD REVIEW ROUND

### REVIEWERS' COMMENTS

### REVIEWER #1

No other comments.

### REVIEWER #2

Dear Editor,

Please find below my comments on the manuscript. Once they are addressed, I consider the article suitable for publication in MIOC.

Lines 185 and 195: Please remove "(using Guo and Thompson exact test)" from line 185 and leave it only in line 195. In addition, kindly revise the paragraph as follows:

"Hardy–Weinberg equilibrium deviations were evaluated in Genepop v4.3 using the Guo (not Gui) and Thompson exact test. The Markov chain procedure was conducted with 10,000 steps, 20 independent replicates, and 5,000 iterations per replicate."

Line 262: It is not accurate to state that the four statistics "consistently" support K = 6. MedMean indicates the strongest structure signal at four clusters, while MaxMean peaks at K = 4, 6, or 7. I agree that selecting K = 6 is reasonable, but the description of the results should accurately reflect these patterns. I would personally report both K = 4 and K = 6, although I leave the final decision to the authors.

Line 284: Please remove "In our study" to avoid redundancy.

