## [Reviewer Report · FIRST REVIEW ROUND - REVIEWERS COMMENTS]

## REVIEWER #1

The manuscript presents an original and relevant study on the genetic structure of *Triatoma brasiliensis* populations in the municipality of Jaguaruana, located in the semi-arid region of Ceará, Brazil. The methodological approach is sound, combining representative sampling in domestic, peridomestic, and sylvatic environments with robust population genetic analyses using microsatellite markers. The topic addressed is of great public health importance, particularly in regions endemic for Chagas disease, as it highlights the complexity of infestation and reinfestation dynamics in areas subjected to varying environmental pressures.

The identification of significant genetic variability among geographically proximate populations and the absence of correlation with physical distance underscore the multifactorial nature of *T. brasiliensis* colonization, with direct implications for the design of surveillance and vector control strategies. The population structure analyses—including AMOVA, dendrograms, and Bayesian inferences (STRUCTURE)—were conducted appropriately. The finding of multiple infestation sources and gene flow between ecotopes suggests that chemical control alone is insufficient to contain this species, especially in peridomestic areas. Genetic data point to deviations from Hardy-Weinberg equilibrium in several samples, indicating an excess of homozygotes and probable inbreeding. These aspects may be related to the local persistence of colonies and recolonization by surviving individuals following control efforts.

However, some aspects deserve attention to enhance the clarity and impact of the article:

The language and writing style need refinement. While the text is understandable, the English contains constructions that would benefit from revision. It is strongly recommended that the manuscript be reviewed by a native English speaker to ensure grammatical accuracy and fluency.

There are references to “Supplement 1” and “Supplement 2,” which appear to be essential for verifying allele sizes and null allele frequencies. These documents were not included in the submitted files. Their absence partially compromises the full assessment of the results’ consistency.

The discussion of insecticide susceptibility is based on “unpublished data.” It would be advisable to either cite the source or clarify that this is a preliminary observation. If these data are in preparation for publication, this should be explicitly stated.

The conclusion could be more concise and actionable. While the manuscript emphasizes the need for locally adapted strategies, it lacks concrete suggestions on how the genetic findings could be used to inform vector control programs.

A brief reflection at the end of the discussion regarding the study’s limitations—such as the limited number of polymorphic loci, the lack of broader temporal comparisons, or challenges in extrapolation—would enrich the critical analysis of the work.

Additionally, the quality of all figures is very poor, making it difficult to properly evaluate the visual data. The images should be revised and replaced with higher-resolution versions that meet the journal’s standards for publication.

Despite these remarks, the manuscript is scientifically sound and provides a meaningful contribution to the field of medical entomology, particularly regarding Chagas disease surveillance and control in rural Brazil. It is a well-designed study that integrates ecological and genetic data with competence.

I recommend acceptance of the manuscript pending minor revisions, particularly concerning language polishing, inclusion of supplementary data, improvement of figure quality, and minor adjustments in the presentation of results and conclusions. I would like to review the revised version before final acceptance.

## REVIEWER #2

Dear Editor,

Please receive my comments regarding the manuscript “The Complexity of the Population Dynamics of *Triatoma brasiliensis* in Rural Northeast, Brazil, Indicated by Genetic Characterization”. Overall, the manuscript presents important results that contribute to the understanding of *Triatoma brasiliensis*, a vector of Chagas disease, particularly concerning its genetic characteristics in northeastern Brazil. However, there are several important issues that the authors must address and correct before the manuscript can be fully accepted for publication in MIOC.

**Major issues**

In the cover letter, the corresponding author highlights an important point that is omitted from the introduction and should be incorporated: the study area has a different landscape compared to previously studied ones. This aspect should also be discussed in the appropriate section.

Although the authors performed several population genetic analyses, the results are poorly discussed. Is there significant population structure or not? Does it vary according to environmental differences with other regions?

Many inferences are based on the NJ tree. However, several of the discussed groups are not statistically supported. Only four significant clusters (with bootstrap support higher than 50%) are observed: CLop33c1p1 + Jen6c1, JDuaWild2 + JDuaWild4, LatR76 + QuixR27, and CLop23p1 + CLop23p2. Additionally, you should specify that the tree was arbitrarily rooted, as no outgroup was used.

The description of the Structure clusters should be moved from the Discussion to the Results section.

Which statistic was used in the Puechmaille method to select the K value? And which was used in Structure Selector? Furthermore, it is difficult to identify the patterns you mention, given the high levels of admixture and the heterogeneity of individuals within DU.

The AMOVA analysis could be improved by incorporating the hierarchical level of location (i.e., location, DU within location, and individuals within DU). You could also consider comparing peridomestic and sylvatic populations.

I disagree with the idea that departures from Hardy-Weinberg equilibrium are solely due to inbreeding. If that were the case, such a pattern should be observed across all loci. However, locus Tb8124 shows departures in nearly all populations, strongly indicating the presence of null alleles. In some populations (e.g., Lat13, CLop33c1p1, JDuaWild5, CLopR69), inbreeding or substructuring may be present, as suggested by departures in three loci.

**Minor issues**

L50-1: ...but vertical transmission has contributed substantially to the spread of this disease to non-endemic areas.

L54: This implies that approximately 83% of infected individuals live in Brazil. Is that accurate?

L57: Replace “four” with “four species”.

L60: Change “Brasilian” to “Brazilian”.

L60-1: Please rephrase; the sentence appears out of context. Consider clarifying how it connects with the previous content.

L62: Do you mean “domicile”?

L68: Delete “with infection”.

L84-6: Please rephrase.

L90: Citing 13 references is excessive. Please select 4–5 representative examples.

L120: Figure 2 could be moved to supplementary material, as the manuscript do not discuss genetic differences between ecotopes.

L 135-6: “it also analyzed frozen insects”, rewrite.

L171: Which statistical test was used to assess Hardy-Weinberg equilibrium?

L193: Rather than “verify,” it would be more accurate to say you “analyzed” or “studied” genetic structure.

L 205: Please correct the grammar in this sentence.

L220-2: Please explain the NAE method more clearly. Are these values global FSTs?

L225-6: Inbreeding and substructuring are distinct phenomena. Additionally, the presence of null alleles should be considered.

L241: You could consider performing a Mantel test within localities to assess isolation by distance at a finer scale.

L263: The migration test does not appear to contribute any significant results to the main discussion. Migration between localities is not addressed elsewhere.

L513: Table II could be moved to the supplementary material.

L534: Table VI could be moved to the supplementary material or omitted. FST values are already provided in Supplementary Table 1, and geographic distances are not relevant as the Mantel test was non-significant.

## AUTHORS’ RESPONSE TO THE REVIEWERS

Dear Reviewers,

We sincerely thank you for your insightful comments and constructive suggestions, which have significantly improved the quality of our manuscript. The majority of the recommendations were incorporated, and the corresponding modifications are highlighted in the revised version, with the exception of the extensive edits resulting from the English language revision.

Detailed, point-by-point responses to each of your comments are provided below. We truly appreciate the time and effort dedicated to reviewing our work.

**REVIEWER COMMENTS:**

**Reviewer: 1**

**The manuscript presents an original and relevant study on the genetic structure of *Triatoma brasiliensis* populations in the municipality of Jaguaruana, located in the semi-arid region of Ceará, Brazil. The methodological approach is sound, combining representative sampling in domestic, peridomestic, and sylvatic environments with robust population genetic analyses using microsatellite markers. The topic addressed is of great public health importance, particularly in regions endemic for Chagas disease, as it highlights the complexity of infestation and reinfestation dynamics in areas subjected to varying environmental pressures. The identification of significant genetic variability among geographically proximate populations and the absence of correlation with physical distance underscore the multifactorial nature of *T. brasiliensis* colonization, with direct implications for the design of surveillance and vector control strategies. The population structure analyses—including AMOVA, dendrograms, and Bayesian inferences (STRUCTURE)—were conducted appropriately. The finding of multiple infestation sources and gene flow between ecotopes suggests that chemical control alone is insufficient to contain this species, especially in peridomestic areas. Genetic data point to deviations from Hardy-Weinberg equilibrium in several samples, indicating an excess of homozygotes and probable inbreeding. These aspects may be related to the local persistence of colonies and recolonization by surviving individuals following control efforts. However, some aspects deserve attention to enhance the clarity and impact of the article:**

**The language and writing style need refinement. While the text is understandable, the English contains constructions that would benefit from revision. It is strongly recommended that the manuscript be reviewed by a native English speaker to ensure grammatical accuracy and fluency.**

The manuscript was revised by a native English speaker.

**There are references to “Supplement 1” and “Supplement 2,” which appear to be essential for verifying allele sizes and null allele frequencies. These documents were not included in the submitted files. Their absence partially compromises the full assessment of the results’ consistency.**

The supplementary materials were previously provided as appendices.

**The discussion of insecticide susceptibility is based on “unpublished data.” It would be advisable to either cite the source or clarify that this is a preliminary observation. If these data are in preparation for publication, this should be explicitly stated.**

This data was generated by REMOT, and this information has now been incorporated into the manuscript.

**The conclusion could be more concise and actionable. While the manuscript emphasizes the need for locally adapted strategies, it lacks concrete suggestions on how the genetic findings could be used to inform vector control programs.**

**A brief reflection at the end of the discussion regarding the study’s limitations—such as the limited number of polymorphic loci, the lack of broader temporal comparisons, or challenges in extrapolation—would enrich the critical analysis of the work.**

The discussion is indeed relevant; nevertheless, our results indicate that, despite the limited number of markers available, it was still possible to reveal distinct dispersion patterns across different ecological contexts. In our previous study in the municipality of Tauá, the sample exhibited a panmictic pattern, which contrasts with the population subdivision detected in Jaguaruana when using the same molecular markers.

**Additionally, the quality of all figures is very poor, making it difficult to properly evaluate the visual data. The images should be revised and replaced with higher-resolution versions that meet the journal’s standards for publication.**

The figures have also been provided as an appendix, in the resolution required by the journal.

**Despite these remarks, the manuscript is scientifically sound and provides a meaningful contribution to the field of medical entomology, particularly regarding Chagas disease surveillance and control in rural Brazil. It is a well-designed study that integrates ecological and genetic data with competence. I recommend acceptance of the manuscript pending minor revisions, particularly concerning language polishing, inclusion of supplementary data, improvement of figure quality, and minor adjustments in the presentation of results and conclusions. I would like to review the revised version before final acceptance.**

**Reviewer: 2**

**Dear Editor,**

**Please receive my comments regarding the manuscript “The Complexity of the Population Dynamics of *Triatoma brasiliensis* in Rural Northeast, Brazil, Indicated by Genetic Characterization”. Overall, the manuscript presents important results that contribute to the understanding of *Triatoma brasiliensis*, a vector of Chagas disease, particularly concerning its genetic characteristics in northeastern Brazil. However, there are several important issues that the authors must address and correct before the manuscript can be fully accepted for publication in MIOC.**

**Major issues**

**In the cover letter, the corresponding author highlights an important point that is omitted from the introduction and should be incorporated: the study area has a different landscape compared to previously studied ones. This aspect should also be discussed in the appropriate section.**

Done.

**Although the authors performed several population genetic analyses, the results are poorly discussed. Is there significant population structure or not? Does it vary according to environmental differences with other regions?**

The discussion has been appropriately revised, providing greater clarity to the aspect under consideration.

**Many inferences are based on the NJ tree. However, several of the discussed groups are not statistically supported. Only four significant clusters (with bootstrap support higher than 50%) are observed: CLop33c1p1 + Jen6c1, JDuaWild2 + JDuaWild4, LatR76 + QuixR27, and CLop23p1 + CLop23p2. Additionally, you should specify that the tree was arbitrarily rooted, as no outgroup was used.**

The necessary clarifications have been included in the text.

**The description of the Structure clusters should be moved from the Discussion to the Results section.**

Done

**Which statistic was used in the Puechmaille method to select the K value? And which was used in Structure Selector? Furthermore, it is difficult to identify the patterns you mention, given the high levels of admixture and the heterogeneity of individuals within DU.**

Puchamaille’s method uses the values of median of means, maximum of means, median of medians, and maximum of medians. Evanno’s method, which uses only DeltaK values. Both methods are reliable, but each has its own particularities that must be evaluated for its data set. The StructureSelector program has both methods (Evanno and Puchamaille).

**The AMOVA analysis could be improved by incorporating the hierarchical level of location (i.e., location, DU within location, and individuals within DU). You could also consider comparing peridomestic and sylvatic populations.**

We conducted a hierarchical-level AMOVA analysis as suggested; nevertheless, the interpretation of the results is consistent with that already presented in the manuscript. Consequently, we opted not to include these additional analyses in the text.

**I disagree with the idea that departures from Hardy-Weinberg equilibrium are solely due to inbreeding. If that were the case, such a pattern should be observed across all loci. However, locus Tb8124 shows departures in nearly all populations, strongly indicating the presence of null alleles. In some populations (e.g., Lat13, CLop33c1p1, JDuaWild5, CLopR69), inbreeding or substructuring may be present, as suggested by departures in three loci.**

It is true that deviations from Hardy–Weinberg equilibrium may also result from null alleles. This possibility was investigated, and our results indicate that the null alleles present in the sample do not influence the outcomes of the analysis, as reported in the Results section.

**Minor issues**

**L50-1: ...but vertical transmission has contributed substantially to the spread of this disease to non-endemic areas.** Done.

**L54: This implies that approximately 83% of infected individuals live in Brazil. Is that accurate?**

The statement that approximately 83% of infected individuals live in Brazil is not supported by the references used in the manuscript. According to WHO (2018, 2022) and the 2nd Brazilian Consensus on Chagas Disease (2015), it is estimated that around 6 million people are infected worldwide, while Brazil accounts for between 1.9 and 4.6 million of these cases. This corresponds to approximately 32% to 77% of global infections.

**L57: Replace “four” with “four species”.** Done.

**L60: Change “Brasilian” to “Brazilian”.** Done

**L60-1: Please rephrase; the sentence appears out of context. Consider clarifying how it connects with the previous content.** Done

**L62: Do you mean “domicile”?**

The term “domiciliary” is correct as it complements the word “ecotope” in the context of the sentence. However, it can be changed if it improves clarity for the reader.

**L68: Delete “with infection”.** Done

**L84-6: Please rephrase.** Done

**L90: Citing 13 references is excessive. Please select 4–5 representative examples.**

Thank you for your suggestion. While I understand the recommendation to limit the number of references, I believe that citing all 13 studies is important to fully support the statement, as it reflects the diversity and breadth of research conducted on gene flow, vector dispersal, and taxonomic assessment of vectors. Each of these studies contributes unique insights to the topic and including them provides a comprehensive overview.

**L120: Figure 2 could be moved to supplementary material, as the manuscript do not discuss genetic differences between ecotopes.**

The figure was actually not adequate. We changed the figure to demonstrate the differences between the two ecotopes.

**L 135-6: “it also analyzed frozen insects”, rewrite.** Done

**L171: Which statistical test was used to assess Hardy-Weinberg equilibrium?**

The statistical test used by Arlequin to assess Hardy-Weinberg equilibrium is the exact Hardy-Weinberg test, which is based on probability values. Specifically, Arlequin implements the exact test developed by Guo and Thompson (1992), which evaluates whether the observed allele frequencies significantly differ from those expected under Hardy-Weinberg equilibrium. This test is particularly useful for small sample sizes and provides a reliable way to assess genetic equilibrium in populations.

**L193: Rather than “verify,” it would be more accurate to say you “analyzed” or “studied” genetic structure.** Done

**L 205: Please correct the grammar in this sentence.** Done

**L220-2: Please explain the NAE method more clearly. Are these values global FSTs?**

The ENA method is described in Chapuis and Estoup 2007. Basically, it uses the Fst estimate described by Weir (1996) for the entire set of microsatellites, as it allows greater accuracy in estimating these values in the presence of null alleles, and the correction is made by the method described by Cavalli-Sforza and Edwards (1967). Through the use of bootstrap, the lowest and highest values of the 95% confidence intervals are obtained, taking the quantiles of 2.5% and 97.5% respectively. The global Fst (both with null alleles and without null alleles) obtained must include both values for the method to be reliable.

**L225-6: Inbreeding and substructuring are distinct phenomena. Additionally, the presence of null alleles should be considered.**

The presence of null alleles is indeed an important consideration and has been discussed in the Discussion section.

**L241: You could consider performing a Mantel test within localities to assess isolation by distance at a finer scale.**

Fine-scale analyses, including Mantel tests within localities to assess isolation by distance, were performed; however, no significant relationships were detected.

**L263: The migration test does not appear to contribute any significant results to the main discussion. Migration between localities is not addressed elsewhere.**

The migration test results corroborate the other analyses, suggesting low migration among subpopulations. Accordingly, we deemed it important to include these results in the manuscript.

**L513: Table II could be moved to the supplementary material.**

**L534: Table VI could be moved to the supplementary material or omitted. FST values are already provided in Supplementary Table 1, and geographic distances are not relevant as the Mantel test was non-significant.**

Regarding the last comments, we understand your point of view; however, we ask to maintain the format traditionally used in our publications. We consider these tables to contain information that is frequently consulted, especially for studies on the same topic.

---

## [Reviewer Report · REVIEWERS COMMENTS]

## REVIEWER #1

No other comments.

## REVIEWER #2

Please receive my comments regarding the revised version of this manuscript.

To better understand the revisions, in several cases I have copied my original question, the authors’ answers, and my reply, because in many cases my proposals were not properly understood or addressed.

I have also added some new comments.

**Major issues**

In the cover letter, the corresponding author highlights an important point that is omitted from the introduction and should be incorporated: the study area has a different landscape compared to previously studied ones. This aspect should also be discussed in the appropriate section.

Done.

Regarding this topic:

L 81-82. Change “within different natural ecotope of Jaguaruana to “with a different natural ecotope from Jaguaruana”

Many inferences are based on the NJ tree. However, several of the discussed groups are not statistically supported. Only four significant clusters (with bootstrap support higher than 50%) are observed: CLop33c1p1 + Jen6c1, JDuaWild2 + JDuaWild4, LatR76 + QuixR27, and CLop23p1 + CLop23p2. Additionally, you should specify that the tree was arbitrarily rooted, as no outgroup was used.

The necessary clarifications have been included in the text.

L192-193. Delete “this tree lacks statistical support”. It is not correct. Some clusters are statistically supported while others are not and this should be informed in Results. In addition, the authors insist on reporting and discussing the implications of clusters not statistically supported (Results L258–263, Discussion L314–317).

Which statistic was used in the Puechmaille method to select the K value? And which was used in Structure Selector? Furthermore, it is difficult to identify the patterns you mention, given the high levels of admixture and the heterogeneity of individuals within DU.

Puchamaille’s method uses the values of median of means, maximum of means, median of medians, and maximum of medians. Evanno’s method, which uses only DeltaK values. Both methods are reliable, but each has its own particularities that must be evaluated for its data set. The StructureSelector program has both methods (Evanno and Puchamaille).

The authors explained the method in general but did not answer my question: Which statistic(s) did you use? Is there a graphic or p-value that shows why you chose K = 6? If so, please provide it.

The AMOVA analysis could be improved by incorporating the hierarchical level of location (i.e., location, DU within location, and individuals within DU). You could also consider comparing peridomestic and sylvatic populations.

We conducted a hierarchical-level AMOVA analysis as suggested; nevertheless, the interpretation of the results is consistent with that already presented in the manuscript. Consequently, we opted not to include these additional analyses in the text.

It is unfortunate that the authors did not incorporate these results into the manuscript. You have a sampling structure completely suitable for this. AMOVA is a hierarchical test: if you only use the level “among populations” you already have the pairwise Fsts, so a global Fst adds almost no information.

I disagree with the idea that departures from Hardy-Weinberg equilibrium are solely due to inbreeding. If that were the case, such a pattern should be observed across all loci. However, locus Tb8124 shows departures in nearly all populations, strongly indicating the presence of null alleles. In some populations (e.g., Lat13, CLop33c1p1, JDuaWild5, CLopR69), inbreeding or substructuring may be present, as suggested by departures in three loci.

It is true that deviations from Hardy–Weinberg equilibrium may also result from null alleles. This possibility was investigated, and our results indicate that the null alleles present in the sample do not influence the outcomes of the analysis, as reported in the Results section

The authors are mistaken on this point. They tend to mix up Fst with Fis results. The ENA correction was applied to Fst estimates. But my concern was that they did not incorporate the possibility that null alleles explain departures from HWE within populations, which directly affects Fis. Null alleles typically generate an excess of homozygotes, which is indeed observed here. I could not access the null alleles file in Supplement 2.

In addition, please order the Results section logically. Currently:

• Lines 228–233: HW departures

• Lines 234–238: effects of null alleles on pairwise Fst

• Lines 239–242: AMOVA

• Lines 242–254: return to Fis (HW departures)

• Lines 243-254: return to Fst

**Minor issues**

L171: Which statistical test was used to assess Hardy-Weinberg equilibrium?

The statistical test used by Arlequin to assess Hardy-Weinberg equilibrium is the exact Hardy-Weinberg test, which is based on probability values. Specifically, Arlequin implements the exact test developed by Guo and Thompson (1992), which evaluates whether the observed allele frequencies significantly differ from those expected under Hardy-Weinberg equilibrium. This test is particularly useful for small sample sizes and provides a reliable way to assess genetic equilibrium in populations.

You should incorporate this information in the manuscript. A single line stating that you used the Guo and Thompson exact test is sufficient.

L193: Rather than “verify,” it would be more accurate to say you “analyzed” or “studied” genetic structure. Done

This correction was not made

L225-6: Inbreeding and substructuring are distinct phenomena. Additionally, the presence of null alleles should be considered.

The presence of null alleles is indeed an important consideration and has been discussed in the Discussion section.

This is incorrect. The presence of null alleles is not discussed in the Discussion section.

L263: The migration test does not appear to contribute any significant results to the main discussion. Migration between localities is not addressed elsewhere.

The migration test results corroborate the other analyses, suggesting low migration among subpopulations. Accordingly, we deemed it important to include these results in the manuscript.

Despite the insistence on keeping the migration results, not a single word is devoted to discussing them.

**New comments:**

L 85-88. Change the sentence “In the state of…” to “In the state of Paraíba, also in northeastern Brazil, Almeida et al. (19) used the mitochondrial cytb gene and, in contrast to the study described above, suggested that *T. brasiliensis* populations are genetically structured.”

L194: As previously requested, please add the method used to test departures from HWE. You explained this in your response but did not incorporate it into the text.

L209. Delete “from”

L210. Change “Markov Chain and Monte Carlo” to “Monte Carlo Markov Chain”

L211. Ancestral mixture model is not correct. Change “correlated allele frequencies”

L235. This is not the only point regarding null alleles. The issue is that you declare substructuring in the Discussion, when null alleles could be generating the departures from HWE. This possibility should be properly addressed.

L288-289. Or null alleles

**Figures and Tables**

Figure 1. The idea is very good. Could you make it tighter? Panel B is unequally distributed within the black frame.

Figure 2 legend. Specify to which locality each number corresponds.

Table IV. Contains several errors (e.g., a p-value of 0.0314 in Clop17 is not marked as significant; a p-value of 0.0128 in Clop15c2 is also unmarked, and others).

Tables. Seven tables are excessive. Please (again) consider moving some to the supplementary material.

## AUTHORS’ RESPONSE TO THE REVIEWERS

**REVIEWER COMMENTS:**

**Reviewer: 2**

**Please receive my comments regarding the revised version of this manuscript. To be er understand the revisions, in several cases I have copied my original ques on, the authors’ answers, and my reply, because in many cases my proposals were not properly understood or addressed. I have also added some new comments.**

**Major issues**

**In the cover le er, the corresponding author highlights an important point that is omi ed from the introduc on and should be incorporated: the study area has a different landscape compared to previously studied ones. This aspect should also be discussed in the appropriate sec on.**

Done.

**Regarding this topic:**

**L 81-82. Change “within different natural ecotope of Jaguaruana to “with a different natural ecotope from Jaguaruana”** Done.

**Many inferences are based on the NJ tree. However, several of the discussed groups are not sta s cally supported. Only four significant clusters (with bootstrap support higher than 50%) are observed: CLop33c1p1 + Jen6c1, JDuaWild2 + JDuaWild4, LatR76 + QuixR27, and CLop23p1 + CLop23p2. Addi onally, you should specify that the tree was arbitrarily rooted, as no outgroup was used.**

The necessary clarifica ons have been included in the text.

**L192-193. Delete “this tree lacks sta s cal support”. It is not correct. Some clusters are sta s cally supported while others are not and this should be informed in Results.**

This has been corrected.

**In addi on, the authors insist on repor ng and discussing the implica ons of clusters not sta s cally supported (Results L258–263, Discussion L314–317).**

“The result concerning João Duarte (JDuaWild5) and Latadas REMOT (LaR25) was excluded, and only groupings with bootstrap values greater than 50 were retained.”

**Which sta s c was used in the Puechmaille method to select the K value? And which was used in Structure Selector? Furthermore, it is difficult to iden fy the pa erns you men on, given the high levels of admixture and the heterogeneity of individuals within DU.**

**Puchamaille’s method uses the values of median of means, maximum of means, median of medians, and maximum of medians. Evanno’s method, which uses only DeltaK values. Both methods are reliable, but each has its own par culari es that must be evaluated for its data set. The StructureSelector program has both methods (Evanno and Puchamaille).**

**The authors explained the method in general but did not answer my ques on: Which sta s c(s) did you use? Is there a graphic or p-value that shows why you chose K = 6? If so, please provide it.**

The sta s cs used were added to the manuscript text, and the graphs were included as Supplement 3.

**The AMOVA analysis could be improved by incorpora ng the hierarchical level of loca on (i.e., loca on, DU within loca on, and individuals within DU). You could also consider comparing peridomes c and sylva c popula ons.**

**We conducted a hierarchical-level AMOVA analysis as suggested; nevertheless, the interpreta on of the results is consistent with that already presented in the manuscript. Consequently, we opted not to include these addi onal analyses in the text.**

**It is unfortunate that the authors did not incorporate these results into the manuscript. You have a sampling structure completely suitable for this. AMOVA is a hierarchical test: if you only use the level “among popula ons” you already have the pairwise Fsts, so a global Fst adds almost no informa on.**

We appreciate the reviewer’s sugges on. While hierarchical AMOVA is indeed informa ve, our study was designed to assess gene c structure primarily at the DU level, which is directly addressed by the AMOVA and pairwise Fst results already presented. Exploratory hierarchical AMOVA produced pa erns consistent with those reported, without altering the interpreta on. To maintain focus and avoid redundancy, we retained the current analy cal framework, which adequately addresses the study objec ves.

**I disagree with the idea that departures from Hardy-Weinberg equilibrium are solely due to inbreeding. If that were the case, such a pa ern should be observed across all loci. However, locus Tb8124 shows departures in nearly all popula ons, strongly indica ng the presence of null alleles. In some popula ons (e.g., Lat13, CLop33c1p1, JDuaWild5, CLopR69), inbreeding or substructuring may be present, as suggested by departures in three loci.**

**It is true that devia ons from Hardy–Weinberg equilibrium may also result from null alleles. This possibility was inves gated, and our results indicate that the null alleles present in the sample do not influence the outcomes of the analysis, as reported in the Results sec on.**

**The authors are mistaken on this point. They tend to mix up Fst with Fis results. The ENA correc on was applied to Fst esmates. But my concern was that they did not incorporate the possibility that null alleles explain departures from HWE within popula ons, which directly affects Fis. Null alleles typically generate an excess of homozygotes, which is indeed observed here. I could not access the null alleles file in Supplement 2.** Yes, it was indeed confusing.

The text has been reorganized, and I hope it now aligns with your considera ons.

**In addi on, please order the Results sec on logically. Currently:**

**• Lines 228–233: HW departures**

**• Lines 234–238: effects of null alleles on pairwise Fst**

**• Lines 239–242: AMOVA**

**• Lines 242–254: return to Fis (HW departures)**

**• Lines 243-254: return to Fst**

The effects of null alleles were moved to the end of the AMOVA results sec on.

**Minor issues**

**L171: Which sta s cal test was used to assess Hardy-Weinberg equilibrium?**

**The sta s cal test used by Arlequin to assess Hardy-Weinberg equilibrium is the exact Hardy Weinberg test, which is based on probability values. Specifically, Arlequin implements the exact test developed by Guo and Thompson (1992), which evaluates whether the observed allele frequencies significantly differ from those expected under Hardy-Weinberg equilibrium. This test is par cularly useful for small sample sizes and provides a reliable way to assess gene c equilibrium in popula ons.**

**You should incorporate this informa on in the manuscript. A single line sta ng that you used the Guo and Thompson exact test is sufficient.**

Done.

**L193: Rather than “verify,” it would be more accurate to say you “analyzed” or “studied” gene c structure.** Done.

**This correc on was not made** Done.

**L225-6: Inbreeding and substructuring are dis nct phenomena. Addi onally, the presence of null alleles should be considered.**

**The presence of null alleles is indeed an important considera on and has been discussed in the Discussion sec on.**

**This is incorrect. The presence of null alleles is not discussed in the Discussion sec on.**

**L263: The migra on test does not appear to contribute any significant results to the main discussion. Migra on between locali es is not addressed elsewhere.**

**The migra on test results corroborate the other analyses, sugges ng low migra on among subpopula ons. Accordingly, we deemed it important to include these results in the manuscript.**

**Despite the insistence on keeping the migra on results, not a single word is devoted to discussing them.**

Table VII was removed from the manuscript, and only the descrip on of the results remained in the text. Indeed, the migra on test was not included in the discussion; its result was added to support the evidence of low migra on among popula ons.

**New comments:**

**L 85-88. Change the sentence “In the state of…” to “In the state of Paraíba, also in northeastern Brazil, Almeida et al. (19) used the mitochondrial cytb gene and, in contrast to the study described above, suggested that *T. brasiliensis* popula ons are gene cally structured.”**

Done.

**L194: As previously requested, please add the method used to test departures from HWE. You explained this in your response but did not incorporate it into the text.** Done.

**L209. Delete “from”** Done.

**L210. Change “Markov Chain and Monte Carlo” to “Monte Carlo Markov Chain”** Done.

**L211. Ancestral mixture model is not correct. Change “correlated allele frequencies”** Done.

**L235. This is not the only point regarding null alleles. The issue is that you declare substructuring in the Discussion, when null alleles could be genera ng the departures from HWE. This possibility should be properly addressed.** Done.

**L288-289. Or null alleles** Done.

**Figures and Tables**

**Figure 1. The idea is very good. Could you make it ghter? Panel B is unequally distributed within the black frame.** Done.

**Figure 2 legend. Specify to which locality each number corresponds.** The locali es were already specified in the figure legend.

**Table IV. Contains several errors (e.g., a p-value of 0.0314 in Clop17 is not marked as significant; a p-value of 0.0128 in Clop15c2 is also unmarked, and others).**

Significant p-values were only marked in places where they were not originally shown; however, for clarity, we have now added * to all values ≤ 0.05.

**Tables. Seven tables are excessive. Please (again) consider moving some to the supplementary material.**

We have removed Table VII, and we hope the revision is now suitable.

---

## [Reviewer Report · REVIEWERS COMMENTS]

## REVIEWER #1

No other comments.

## REVIEWER #2

Dear Editor,

Please find below my comments on the manuscript. Once they are addressed, I consider the article suitable for publication in MIOC.

Lines 185 and 195: Please remove “(using Guo and Thompson exact test)” from line 185 and leave it only in line 195. In addition, kindly revise the paragraph as follows:

“Hardy–Weinberg equilibrium deviations were evaluated in Genepop v4.3 using the Guo (not Gui) and Thompson exact test. The Markov chain procedure was conducted with 10,000 steps, 20 independent replicates, and 5,000 iterations per replicate.”

Line 262: It is not accurate to state that the four statistics “consistently” support K = 6. MedMean indicates the strongest structure signal at four clusters, while MaxMean peaks at K = 4, 6, or 7. I agree that selecting K = 6 is reasonable, but the description of the results should accurately reflect these patterns.

I would personally report both K = 4 and K = 6, although I leave the final decision to the authors.

Line 284: Please remove “In our study” to avoid redundancy.